# Monotonic Variational Gaussian Process for Efficient Data Collection

Donghyun Lee [1]    Young Myoung Ko [1]

## Abstract

Modeling the learning curve is critical for cost-effective data collection in deep learning systems. Most prior approaches assume a specific parametric learning curve, but these can be inappropriate when no reliable parametric form can be assumed for the learning curve. While Gaussian processes offer flexible nonparametric modeling, existing GP approaches that enforce monotonicity typically introduce intractable factors or require derivative observations. To address this, we propose a Monotonic Variational Gaussian Process for Efficient Data Collection (MOVE), which (i) introduces a novel monotonic variational GP formulation with virtual-derivative factors to enable tractable posterior inference, and (ii) develops an expected shortfall based objective for target-driven data collection. Furthermore, our theoretical analysis shows that expected shortfall provides non-vanishing gradient signals that enable reliable gradient-based optimization. Extensive experiments on classification, segmentation, and detection benchmarks demonstrate consistent improvements over the prior method.

## 1. Introduction

When training deep learning models, larger training sets are empirically observed to improve model performance (Hestness et al., 2017), motivating the use of large standard datasets. However, these datasets often do not match the target domain in real-world applications (Koh et al., 2021). This limitation leads practitioners to collect additional task-specific data. Since both data collection and training incur costs, it becomes essential to decide how much additional data to acquire and from which sources.

Prior research has explored how to use limited data collection and computational budgets more efficiently from several angles, including extracting informative data from data streams (Gomes & Krause, 2010; Kim et al., 2021), analyzing learning curves to relate performance to dataset size (Hoiem et al., 2021; Loureiro et al., 2021), and optimizing data collection or resource allocation by fitting power law curves (Mahmood et al., 2022; Wang et al., 2024). Learning curve based optimization typically assumes a specific parametric relationship between performance and training set size. While such models are simple and effective when the assumptions hold, they are difficult to apply in multi-source data collection settings where no reliable parametric learning curve can be assumed and the performance response to each source is unknown. Therefore, we adopt a Gaussian process as a nonparametric model to capture unknown performance responses across multiple data sources.

For example, consider a company deploying a vision model across multiple factories, where each site exhibits a different domain shift from standard benchmarks. Improving performance requires collecting additional labeled images from multiple factories, but annotation is expensive. Collecting too few labels risks missing the target, while collecting too many wastes budget. With unknown and noisy source-wise responses, the practitioner needs a method to decide how many additional labels to collect from each factory to reach a target at minimal cost, assuming that collecting more labeled data from any factory should not worsen performance.

The structural prior knowledge for data collection is that acquiring additional data from any source should not reduce performance. Monotonicity is commonly enforced using derivative constraints and derivative-based GP models often assume derivative observations or reliable gradients (Solak et al., 2002; Padidar et al., 2021). However, each evaluation provides only a performance value rather than a slope, so estimating gradients requires many controlled repeats that change one source at a time. Constrained GP methods avoid requiring derivative observations by embedding operator constraints into the prior (Jidling et al., 2017; Lange-Hegermann, 2018; Besginow & Lange-Hegermann, 2022). But these methods typically impose hard constraints and can be fragile when learning curve measurements are noisy. In practice, noisy training can create locally non-monotone regions in the observed learning curve. In addition, stable gradient-based objectives for data collection under Gaussian

[1]Department of Industrial and Management Engineering, Pohang University of Science and Technology, Pohang, Republic of Korea. Correspondence to: Young Myoung Ko <youngko@postech.ac.kr>.

*Proceedings of the 43$^{rd}$ International Conference on Machine Learning*, Seoul, South Korea. PMLR 306, 2026. Copyright 2026 by the author(s).

processes remain underdeveloped.

To address these limitations, we introduce MOVE, which couples a monotonic variational learning curve model with a decision objective designed for effective data collection optimization. MOVE consists of two main components (i) *Learning curve prediction*: We propose a monotonic variational Gaussian process enforcing coordinate-wise monotonicity via probabilistic virtual derivatives. (ii) *Data collection optimization*: Based on the posterior predictive distribution derived by our monotonic variational model, we design an expected shortfall objective. And we provide a theoretical justification that it produces more stable gradient signals by penalizing the expected shortfall below the target.

### 1.1. Related Work

**Gaussian processes and monotonic function modeling.** Gaussian process (GP) regression is a nonparametric model that can fit an unknown function and quantify predictive uncertainty, and it is widely used in probabilistic machine learning (Williams & Rasmussen, 2006).

Monotonic function modeling has also been studied through a range of shape constrained and Bayesian nonparametric approaches, including methods that impose monotonicity via priors (Tibshirani et al., 2011; Westling & Carone, 2020). Within the GP literature, monotonicity has been enforced using projection based methods that map GP samples onto monotonic function spaces (Lin & Dunson, 2014), operator-based constraints (Agrell, 2019), and virtual derivative constraints (Riihimäki & Vehtari, 2010). These approaches primarily target shape-constrained prediction. In contrast, data collection optimization uses the predictive distribution to compute an objective, requiring inference that remains tractable and is guided by a well-defined objective.

**Learning curves and dataset size estimation.** The relationship between model performance and training set size is often studied through learning curves and neural scaling laws. Empirical work has shown that deep networks frequently exhibit approximate power law scaling with dataset size and model capacity in certain regimes (Hestness et al., 2017; Kaplan et al., 2020), and subsequent studies have refined these observations for modern architectures and tasks (Loureiro et al., 2021; Alabdulmohsin et al., 2022).

However, later works also point out important limitations of such parametric scaling laws. The apparent power law behavior often holds over a restricted range of dataset sizes or losses (Hoiem et al., 2021; Jain et al., 2023). In addition, prior work has noted that learning curve models relying on restrictive parametric models can extrapolate poorly when real learning curves deviate from assumed forms (Rakotoarison et al., 2024). These observations motivate more flexible, nonparametric approaches to learning curve modeling that

do not rely on a single fixed functional form.

**Subset selection and data collection.** The high cost of data annotation and the limited availability of computational resources make it unrealistic to collect or train on all useful data. This challenge has motivated extensive work on improving data efficiency and model training. Budgeted nonparametric learning from data streams maintains only a small set of informative data under memory constraints, often using submodular selection rules (Gomes & Krause, 2010). Coreset and subset selection methods construct small training subsets that approximate learning on the full dataset and have been applied to classical models and deep networks (Mirzasoleiman et al., 2020; Borsos et al., 2020; Killamsetty et al., 2021; Yang et al., 2024). Other approaches reduce training cost by summarizing a large dataset into a small set of synthetic data, using bilevel optimization to keep accuracy as high as possible (Loo et al., 2023; Maalouf et al., 2023).

While the above methods assume a fixed dataset and focus on training efficiency, adaptive data collection and resource allocation aim to decide how much data to collect next and how to allocate limited budgets accordingly. Some approaches fit parametric learning curves and use these models to decide how much additional data or computation is needed to meet performance or success thresholds (Mahmood et al., 2022; Wang et al., 2024). Other work studies budgeted data collection in structured settings, such as allocating a fixed annotation budget across multiple distributions (Zang et al., 2025), or choosing between expensive strong labels and cheaper weak labels under an overall budget (Tejero et al., 2023). These parametric learning curve studies typically rely on a specific parametric form. In contrast, we propose a monotonic variational Gaussian process and optimize data collection decisions with an expected shortfall objective that directly penalizes under-target shortfall and yields non-vanishing gradients below the target.

## 2. Problem Formulation

In this section, we introduce the notation and formulate a standard data collection problem used in prior work, followed by a discussion of its limitations.

**Notations.** We consider a multi-source setting with $K$ domains. Source $k \in \{1, \ldots, K\}$ corresponds to an unknown data distribution $\mathcal{P}_k$ over an input space $\mathbb{N}^k$, and collecting $n_k$ labeled data from source $k$ yields a labeled dataset of size $\mathbf{n} = (n^1, \ldots, n^K)^\top$. Given such a dataset, we write $V(\mathbf{n})$ for a scalar performance metric obtained by training a model on $\mathbf{n}$.

Data are acquired over $T$ decision stages $t \in \{1, \ldots, T\}$.

At each stage $t$ we choose a cumulative labeled-count vector

$$\mathbf{n}_t \in \mathbb{N}^K, \tag{1}$$

starting from an initial state $\mathbf{n}_0 \in \mathbb{N}^K$ and enforcing coordinate-wise monotonicity constraints.

$$\mathbf{n}_0 \leq \mathbf{n}_1 \leq \cdots \leq \mathbf{n}_T \tag{2}$$

The increment collected at stage $t$ is implicitly given by $\mathbf{n}_t - \mathbf{n}_{t-1}$. The sequence

$$(\mathbf{n}_1, \ldots, \mathbf{n}_T) \tag{3}$$

fully specifies a data collection plan, and we denote by $\mathcal{F}$ the set of feasible decisions induced by these nonnegativity, integrality, and monotonicity constraints.

Consider the case in Section 1, where factory $k$ corresponds to a data source $\mathcal{P}_k$ producing inspection images. At stage $t$, the vector $\mathbf{n}_t$ specifies the cumulative number of labeled images to be collected from each factory, and $\mathbf{n}_T$ denotes the total number of labeled images collected per factory by the end of the planning horizon. Given a target performance level $V^*$, we choose a sequence $(\mathbf{n}_1, \ldots, \mathbf{n}_T) \in \mathcal{F}$ such that the final state $\mathbf{n}_T$ is sufficient to reach $V^*$ while minimizing the overall acquisition cost, summarized by $\mathcal{L}(\mathbf{n}_1, \ldots, \mathbf{n}_T)$.

### 2.1. Problem Statement

Following the target-driven formulation of Mahmood et al. (2022), we model performance in terms of a success probability, which we estimate using a learning curve model. We define

$$F(\mathbf{n}) := \mathbb{P}\big[V(\mathbf{n}) \geq V^*\big] \in [0, 1]. \tag{4}$$

as the probability that training on $\mathbf{n}$ attains at least the target performance. At stage $t$, the increment $\mathbf{n}_t - \mathbf{n}_{t-1}$ specifies how much additional labeled data is collected from each source, and a fixed cost vector $\mathbf{c} \in \mathbb{R}_+^K$ assigns a per-sample labeling cost to each source.

We model the data collection problem by defining the following objective for a decision $(\mathbf{n}_1, \ldots, \mathbf{n}_T)$

$$
\begin{aligned}
\mathcal{L}(\mathbf{n}_1, \ldots, \mathbf{n}_T) = &\sum_{t=1}^{T} \mathbf{c}^\top (\mathbf{n}_t - \mathbf{n}_{t-1})\big(1 - F(\mathbf{n}_{t-1})\big) \\
&+ \gamma\big(1 - F(\mathbf{n}_T)\big).
\end{aligned} \tag{5}
$$

where $\gamma > 0$ penalizes failing to reach the target by the end of the horizon. Intuitively, the factor $1 - F(\mathbf{n}_{t-1})$ represents the probability that the target has not been reached before stage $t$.

Therefore, the data collection optimization problem is typically formulated as choosing an increasing sequence of cumulative dataset sizes that minimizes the total loss.

$$
\begin{aligned}
\min_{\mathbf{n}_1, \ldots, \mathbf{n}_T} \quad & \mathcal{L}(\mathbf{n}_1, \ldots, \mathbf{n}_T) \\
\text{s.t.} \quad & \mathbf{n}_0 \leq \mathbf{n}_1 \leq \cdots \leq \mathbf{n}_T,
\end{aligned} \tag{6}
$$

### 2.2. Key Challenge

The key challenge in (6) is that decision quality depends critically on how the success model $F(\mathbf{n})$ is constructed and how the penalty term is chosen. Prior work typically defines $F$ by fitting a simple parametric learning curve, which assumes a fixed functional form. In multi-source settings, learning behavior can differ across sources, so a single parametric form can be restrictive (Jiang et al., 2025). Even on standard benchmarks such as CIFAR-10, simple scaling laws often fit only over limited ranges (Sorscher et al., 2022), and this fixed form can distort how additional data from each source affects performance.

The choice of penalty term also influences how effectively the objective guides data collection decisions. By defining a penalty as a function of the terminal shortfall relative to the target, it controls the trade-off between acquisition cost and the risk of missing the target. In addition, because the objective relies on the predictive distribution, the learning curve model must admit tractable inference with stable posterior updates. These considerations motivate a nonparametric monotonic model for $F(\mathbf{n})$ together with an objective that provides informative signals. MOVE addresses these requirements with a monotonic variational GP and an expected shortfall penalty for target-driven data collection decisions.

## 3. Our Approach

We present MOVE, a data collection optimization method that combines a monotonic variational GP learning curve model with an expected-shortfall objective. MOVE predicts how performance improves with additional data from each source and decides how much data to collect across sources to achieve the target performance at low cost.

In Section 3.1, we introduce our monotonic variational Gaussian process architecture, which models the multi-source learning curve as a monotonic function of per-source sample sizes and produces predictive means and variances. In Section 3.2, we define the expected shortfall objective for data collection decision and analyze its optimization signal, showing that expected shortfall provides non-vanishing gradients in the under-target regime.

### 3.1. Learning Curve Prediction

We consider a multi-source learning curve that maps per-source labeled sample counts to task performance. We model this effect through a latent learning curve $f : \mathbb{R}^K \to \mathbb{R}$, where $f(\mathbf{n}) := \mathbb{E}[V(\mathbf{n})]$ denotes the expected task performance. We model $f$ as a smooth function over $\mathbb{R}^K$; during data collection optimization, we optimize a continuous relaxation and enforce feasibility using projected gradient descent.

MOVE approximates this learning curve with a Gaussian process prior

$$f \sim \mathcal{GP}(\mu_0, k), \tag{7}$$

where $\mu_0$ is a learnable constant mean and $k$ is a covariance function. Given $N$ training runs $\mathcal{D} = \{(\mathbf{x}_i, y_i)\}_{i=1}^N$, with $\mathbf{x}_i$ and noisy performance measurements $y_i \approx V(\mathbf{x}_i)$. We assume conditionally independent Gaussian observations

$$y_i = f(\mathbf{x}_i) + \varepsilon_i, \quad \varepsilon_i \sim \mathcal{N}(0, \sigma^2). \tag{8}$$

or, in vector form,

$$\mathbf{y} \mid \mathbf{f} \sim \mathcal{N}(\mathbf{f}, \sigma^2 I). \tag{9}$$

This GP prior provides a flexible nonparametric model for the multi-source learning curve. We incorporate the prior knowledge that acquiring additional labeled data should not reduce performance. To encode this monotonicity, we place virtual derivative constraints at a set of virtual locations $\{\tilde{\mathbf{x}}_j\}_{j=1}^J$ and for each coordinate $k$. For every location $j$ and coordinate $k \in \{1, \dots, K\}$ we define the partial derivative

$$g_{j,k} := \frac{\partial f}{\partial x_k}(\tilde{\mathbf{x}}_j), \tag{10}$$

introduce binary virtual variables $s_{j,k} \in \{0, 1\}$ linked to latent derivatives via a probit likelihood, a widely used construction for relating latent variables to the binary event (Hernández-Lobato et al., 2014). Following the standard virtual-derivative construction (Riihimäki & Vehtari, 2010), we impose monotonicity by introducing latent derivative variables. We apply this construction coordinate-wise to encode monotonicity across all $K$ sources.

$$p(s_{j,k} = 1 \mid g_{j,k}) = \Phi(g_{j,k}), \tag{11}$$

where $\Phi$ is the standard normal CDF. We use this probabilistic formulation to impose monotonicity as a soft constraint, which accommodates uncertainty in the derivatives. In MOVE, we fix $s_{j,k} = 1$ for all $(j, k)$, so these virtual factors act as soft coordinate-wise monotonicity constraints. To control their influence, we use a weighted virtual likelihood with a factor $\lambda > 0$

$$p(\mathbf{s} \mid \mathbf{g}) \propto \prod_{j=1}^J \prod_{k=1}^K \Phi(g_{j,k})^\lambda. \tag{12}$$

Note that, letting $d_{j,k} = e_k$ denote the $k$-th canonical basis vector in $\mathbb{R}^K$, we have $g_{j,k} = d_{j,k}^\top \nabla f(\tilde{\mathbf{x}}_j)$. Because directional derivatives are linear functionals of a GP, the collection of function values and derivatives admits a joint Gaussian prior.

For compact notation, we stack the training function values and virtual derivatives into vectors

$$\begin{aligned} \mathbf{f} &:= \big(f(\mathbf{x}_1), \dots, f(\mathbf{x}_N)\big)^\top \in \mathbb{R}^N, \\ \mathbf{g} &:= \big(g_{j,k}\big)_{j=1,\dots,J;\, k=1,\dots,K} \in \mathbb{R}^{M_g}, \end{aligned} \tag{13}$$

with $M_g := J \cdot K$. Since directional derivatives are linear functionals of $f$, all cross-covariances between function values and directional derivatives follow directly by differentiating $k$ with respect to its inputs. Together with $k(\mathbf{x}, \mathbf{x}') = \text{Cov}\big(f(\mathbf{x}), f(\mathbf{x}')\big)$, these expressions fully determine the block covariance matrix $K_{\text{joint}}$ of the joint Gaussian prior $(\mathbf{f}, \mathbf{g}) \sim \mathcal{N}(\mu_{\text{joint}}, K_{\text{joint}})$. Conditioned on the function values $\mathbf{f}$ and directional derivatives $\mathbf{g}$, the likelihood factorizes as

$$\begin{aligned} p(\mathbf{y}, \mathbf{s} \mid \mathbf{f}, \mathbf{g}) &\propto \prod_{i=1}^N \mathcal{N}\big(y_i \mid f(\mathbf{x}_i), \sigma^2\big) \\ &\cdot \prod_{j=1}^J \prod_{k=1}^K p(s_{j,k} \mid g_{j,k})^\lambda. \end{aligned} \tag{14}$$

Since the probit likelihood term is not Gaussian, the posterior $p(\mathbf{f}, \mathbf{g} \mid \mathbf{y}, \mathbf{s})$ is analytically intractable. However, this formulation is essential in MOVE. It imposes the sign constraint $g_{j,k} \geq 0$ without requiring derivative measurements or committing to any particular derivative magnitude.

To address the non-Gaussian probit likelihood, MOVE adopts a stochastic variational GP, which provides a stable and well-defined posterior approximation in the presence of latent virtual derivative factors. We use a single set of inducing locations $Z = \{\mathbf{z}_r\}_{r=1}^M$ in the learning curve domain. At each location $\mathbf{z}_r$, we introduce one function-value inducing variable and $K$ derivative inducing variables, one per coordinate. In our setting, derivative values are not observed. They enter only through auxiliary probit factors that enforce monotonicity. MOVE integrates these latent virtual-derivative factors by deriving a stochastic variational objective that couples function and derivative inducing variables under the virtual monotonicity constraints.

Collecting the function values as

$$\mathbf{u}_f = \mathbf{f}(Z) = \big(f(\mathbf{z}_1), \dots, f(\mathbf{z}_M)\big)^\top \tag{15}$$

and stacking all directional derivatives

$$\mathbf{u}_g = \big(e_k^\top \nabla f(\mathbf{z}_r)\big)_{r=1,\dots,M;\, k=1,\dots,K} \tag{16}$$

We define the full inducing vector

$$\mathbf{u} = \begin{pmatrix} \mathbf{u}_f \\ \mathbf{u}_g \end{pmatrix} \in \mathbb{R}^{M_u}, \qquad M_u := M(1 + K). \tag{17}$$

Under the GP prior $f \sim \mathcal{GP}(\mu_0, k)$, the inducing vector $\mathbf{u}$ has a multivariate Gaussian prior

$$\mathbf{u} \sim \mathcal{N}(\boldsymbol{\mu}_{\mathbf{u}}, K_{\mathbf{uu}}), \tag{18}$$

where $\boldsymbol{\mu}_{\mathbf{u}}$ collects the prior means of the function values and directional derivatives at $Z$, $K_{\mathbf{uu}}$ is the corresponding block

covariance matrix. Conditioning on the inducing variables, the joint prior factorizes as $p(\mathbf{f}, \mathbf{g}, \mathbf{u}) = p(\mathbf{u}) \, p(\mathbf{f}, \mathbf{g} \mid \mathbf{u})$, where $p(\mathbf{f}, \mathbf{g} \mid \mathbf{u})$ is the exact GP conditional. We choose a Gaussian variational posterior $q(\mathbf{u}) = \mathcal{N}(\mathbf{m}, S)$ and define

$$q(\mathbf{f}, \mathbf{g}, \mathbf{u}) = p(\mathbf{f}, \mathbf{g} \mid \mathbf{u}) \, q(\mathbf{u}), \qquad (19)$$

so that the marginals $q(\mathbf{f})$ and $q(\mathbf{g})$ retain exact GP conditional structure while all trainable parameters are concentrated in $(\mathbf{m}, S)$ and the kernel hyperparameters. The joint model

$$p(\mathbf{y}, \mathbf{s}, \mathbf{f}, \mathbf{g}, \mathbf{u}) = p(\mathbf{u}) p(\mathbf{f}, \mathbf{g} \mid \mathbf{u}) p(\mathbf{y} \mid \mathbf{f}) p(\mathbf{s} \mid \mathbf{g}) \quad (20)$$

induces the marginal likelihood

$$\log p(\mathbf{y}, \mathbf{s}) = \log \int p(\mathbf{y}, \mathbf{s}, \mathbf{f}, \mathbf{g}, \mathbf{u}) \, d\mathbf{f} \, d\mathbf{g} \, d\mathbf{u}. \qquad (21)$$

With this variational posterior, the evidence lower bound (ELBO) becomes

$$\begin{aligned}
ELBO = &\sum_{i=1}^{N} \mathbb{E}_{q(f(\mathbf{x}_i))} \left[ \log \mathcal{N}(y_i \mid f(\mathbf{x}_i), \sigma^2) \right] \\
&+ \lambda \sum_{j=1}^{J} \sum_{k=1}^{K} \mathbb{E}_{q(g_{j,k})} \left[ \log p(s_{j,k} \mid g_{j,k}) \right] \\
&- \mathrm{KL}\big( q(\mathbf{u}) \,\|\, p(\mathbf{u}) \big)
\end{aligned} \qquad (22)$$

and $\log p(s_{j,k} = 1 \mid g_{j,k}) = \log \Phi(g_{j,k})$. Importantly, each monotonicity factor depends only on the one-dimensional marginal $q(g_{j,k})$, and we estimate $\mathbb{E}_{q(g_{j,k})}[\log \Phi(g_{j,k})]$ by Monte Carlo sampling.

After maximizing the ELBO with respect to $q(\mathbf{u})$ and the kernel hyperparameters, we obtain $q(\mathbf{u}) = \mathcal{N}(\mathbf{m}, S)$. For any $\mathbf{n} \in \mathbb{N}^K$ the predictive marginal of the latent learning curve is

$$q\big(f(\mathbf{n})\big) = \mathcal{N}\big(\hat{\mu}(\mathbf{n}), \hat{\sigma}^2(\mathbf{n})\big), \qquad (23)$$

where

$$\begin{aligned}
\hat{\mu}(\mathbf{n}) &= \mu_0 + K_{\mathbf{nu}} K_{\mathbf{uu}}^{-1} \big(\mathbf{m} - \boldsymbol{\mu}_{\mathbf{u}}\big), \\
\hat{\sigma}^2(\mathbf{n}) &= K_{\mathbf{nn}} + K_{\mathbf{nu}} K_{\mathbf{uu}}^{-1} (S - K_{\mathbf{uu}}) K_{\mathbf{uu}}^{-1} K_{\mathbf{un}}.
\end{aligned} \qquad (24)$$

and

$$K_{\mathbf{nu}} := \mathrm{Cov}\big(f(\mathbf{n}), \mathbf{u}\big), \qquad K_{\mathbf{uu}} := \mathrm{Cov}(\mathbf{u}, \mathbf{u}). \quad (25)$$

A detailed derivation is provided in Appendix A.1. After optimizing the ELBO, we obtain closed form predictive mean and variance $\hat{\mu}(\mathbf{n})$ and $\hat{\sigma}^2(\mathbf{n})$ for the learning curve model. In Section 3.2, we use these predictions to define our expected-shortfall objective for multi-source data collection.

### 3.2. Data Collection Optimization

In this section, we show how the predictive distribution $q(f(\mathbf{n}))$ defines a data collection objective for multi-source decision making. We also explain why this objective differs from the failure-probability penalties $\gamma(1 - F(\mathbf{n}_T))$ used in prior work.

We follow the same finite-horizon planning setup and notation as in Section 2. A data collection plan is an increasing sequence $(\mathbf{n}_1, \dots, \mathbf{n}_T) \in \mathcal{F}$ with increments $\mathbf{n}_t - \mathbf{n}_{t-1}$. Given any cumulative state $\mathbf{n} \in \mathbb{N}^K$, MOVE provides a Gaussian predictive distribution for model performance with mean $\hat{\mu}(\mathbf{n})$ and standard deviation $\hat{\sigma}(\mathbf{n})$. For simplicity, we write $\mu(\mathbf{n}) := \hat{\mu}(\mathbf{n})$ and $\sigma(\mathbf{n}) := \hat{\sigma}(\mathbf{n})$. This enables us to express target score attainment through the standardized margin

$$\alpha(\mathbf{n}) = \frac{V^* - \mu(\mathbf{n})}{\sigma(\mathbf{n})}, \qquad (26)$$

and define

$$R(\mathbf{n}) = \Phi\big(\alpha(\mathbf{n})\big), \qquad (27)$$

where $\Phi$ denotes the standard normal CDF. Under the Gaussian model, we estimate the failure probability $1 - F(\mathbf{n})$ by $R(\mathbf{n}) = \mathrm{Pr}_q\big(f(\mathbf{n}) < V^*\big)$.

While $R(\mathbf{n})$ captures how likely we are to miss the target, it does not reflect how far below the target we are when failure occurs. We therefore consider the expected shortfall,

$$\begin{aligned}
S(\mathbf{n}) &= \mathbb{E}_q\big[(V^* - f(\mathbf{n}))_+\big] \\
&= (V^* - \mu(\mathbf{n})) \, \Phi\big(\alpha(\mathbf{n})\big) + \sigma(\mathbf{n}) \, \varphi\big(\alpha(\mathbf{n})\big),
\end{aligned} \qquad (28)$$

where $\varphi$ denotes the standard normal PDF. MOVE combines this quantity into the objective:

$$\begin{aligned}
\mathcal{L}(\mathbf{n}_1, \dots, \mathbf{n}_T) = &\sum_{t=1}^{T} \mathbf{c}_t^\top \big(\mathbf{n}_t - \mathbf{n}_{t-1}\big) R\big(\mathbf{n}_{t-1}\big) \\
&+ \gamma S\big(\mathbf{n}_T\big),
\end{aligned} \qquad (29)$$

where $\mathbf{c}_t$ encodes per-source labeling costs and $\gamma > 0$ controls the strength of the terminal penalty. The shortfall term $S(\mathbf{n}_T)$ replaces the failure probability $(1 - F(\mathbf{n}_T))$. To analyze different penalty choices, we formalize regularity assumptions from the compact decision domain and the MOVE predictive model, together with a mild non-degeneracy condition on predictive uncertainty. In practice, GP implementations routinely add a small variance floor for numerical stability, which makes $\sigma(\cdot)$ strictly positive on $\mathcal{F}$ and hence bounded away from zero.

**Assumption 3.1** (Compact domain). The feasible set $\mathcal{F}$ is nonempty and compact.

**Assumption 3.2** (Differentiability). $\mu, \sigma \in C^2$ on $\mathcal{F}$ and all first and second order derivatives of $\mu, \sigma$ are bounded.

**Assumption 3.3** (Lower bound of variance). There exists $\sigma_{\min} > 0$ such that

$$\sigma(\mathbf{n}) \geq \sigma_{\min} \qquad \text{for all } \mathbf{n} \in \mathcal{F}.$$

We now compare two terminal penalties under the same Gaussian model. The failure probability penalty used in prior work and MOVE's expected-shortfall penalty. If we keep the failure probability, the terminal term becomes $\gamma R(\mathbf{n}_T)$. The following theorems show that this probability penalty suffers from exponentially vanishing gradients, which motivates our replacement by the expected shortfall $S(\mathbf{n}_T)$ in (29).

**Theorem 3.4.** *Suppose Assumption 3.2 and 3.3 hold. Then*

$$\nabla R(\mathbf{n}) = \varphi\big(\alpha(\mathbf{n})\big)\, \nabla \alpha(\mathbf{n}),$$

*and there exist constants $C_1, C_2 > 0$ such that*

$$\|\nabla R(\mathbf{n})\| \leq C_1 \, \varphi\big(\alpha(\mathbf{n})\big)\, (1 + C_2|\alpha(\mathbf{n})|)$$
$$= \frac{C_1}{\sqrt{2\pi}}\, (1 + C_2|\alpha(\mathbf{n})|)\, \exp\big(-\alpha(\mathbf{n})^2/2\big).$$

*Consequently, $\|\nabla R(\mathbf{n})\|$ decays at least exponentially fast in $|\alpha(\mathbf{n})|$ as $|\alpha(\mathbf{n})| \to \infty$.*

Theorem 3.4 formalizes a practical issue we observe empirically. When the model is very confident that $V(\mathbf{n})$ lies far below or far above $V^*$, the probability penalty $R(\mathbf{n})$ becomes nearly flat and provides little gradient signal for the data collection decision. In contrast, the expected shortfall penalty $S(\mathbf{n})$ used in MOVE accounts for both the probability of falling below the target and the magnitude of the resulting shortfall, which avoids the vanishing gradient behavior. The following theorem establishes this key result.

**Theorem 3.5.** *Suppose Assumption 3.2 holds. Then*

$$\nabla S(\mathbf{n}) = -\Phi\big(\alpha(\mathbf{n})\big)\, \nabla \mu(\mathbf{n}) + \varphi\big(\alpha(\mathbf{n})\big)\, \nabla \sigma(\mathbf{n}),$$

*Consequently, $\nabla S(\mathbf{n})$ approaches $-\nabla\mu(\mathbf{n})$ as $\alpha(\mathbf{n}) \to \infty$.*

The proofs of Theorems 3.4 and 3.5 are provided in Appendices A.2 and A.3. These theorems explain the design of (29): MOVE preserves the target-driven structure of the original formulation through $R(\mathbf{n})$ and the terminal term $S(\mathbf{n}_T)$, while replacing a failure-probability penalty with an expected shortfall penalty. This penalty term provides a meaningful gradient signal in under-target regimes and naturally diminishes once the target is safely achieved, leading to more reliable gradient-based multi-source data collection.

For example, if $\mu(\mathbf{n}) \ll V^*$ and $\sigma(\mathbf{n})$ is small, then $\alpha(\mathbf{n}) \gg 0$ and the probability penalty satisfies $R(\mathbf{n}) \approx 1$ with $\nabla R(\mathbf{n}) \approx 0$, so the planner receives almost no incentive to collect more data. In the same regime, however, the

expected shortfall satisfies $\nabla S(\mathbf{n}) \approx -\nabla\mu(\mathbf{n})$, so MOVE continues to collect more data until the predictive mean approaches the target.

We now consider the full multi-stage decision problem and study the smoothness of the MOVE objective with respect to the stage-wise decisions. The following proposition shows that the resulting objective is well-behaved on the feasible set.

**Proposition 3.6.** *Suppose Assumptions 3.1, 3.2 and 3.3 hold. Then the objective $\mathcal{L}$ in (29) is continuously differentiable on $\mathcal{F}$ and has a Lipschitz-continuous gradient on $\mathcal{F}$; There exists $C > 0$ such that*

$$\|\nabla\mathcal{L}(\mathbf{n}_1) - \nabla\mathcal{L}(\mathbf{n}_2)\| \leq C\, \|\mathbf{n}_1 - \mathbf{n}_2\| \quad \text{for all } \mathbf{n}_1, \mathbf{n}_2 \in \mathcal{F}.$$

Proposition 3.6 shows that the MOVE objective is well behaved on the feasible set, and the proof is provided in Appendix A.4. This property ensures that gradient-based decision over the continuous relaxation of the data collection variables admits stable optimization signals, which is essential for reliably solving the finite-horizon decision problem in practice.

As a result, MOVE provides a nonparametric learning curve model that respects performance improvement with additional data and maintains informative optimization signals even when far below the target, enabling stable gradient-based decisions with well-behaved guarantees on the feasible set. MOVE adapts to unknown and potentially complex performance responses across multiple data sources, making it particularly suitable for multi source settings where each source may exhibit different scaling behavior.

## 4. Experiments

In this section, we present our main empirical results on benchmark datasets and analyze the optimization behavior of MOVE. We first compare MOVE with LOC and ExactGP on multi-source vision tasks, reporting cost and accuracy ratios at fixed target performance levels, and additionally present target-sweep curves to illustrate how performance evolves across different target scores. Next, we empirically validate our theorem by contrasting the gradient signals induced by the failure-probability and expected-shortfall penalties, and by tracking how the predicted success probability evolves along the optimization trajectory. We then ablate the learning curve model by replacing our monotonic variational GP with standard SVGP (Hensman et al., 2013) and directional-derivative SVGP (DSVGP) (Padidar et al., 2021), isolating the effect of monotonicity and virtual derivative constraints on predictive accuracy. Data collection descriptions and experimental setup are deferred to Appendix B.

*Table 1.* Performance comparison between LOC, ExactGP, and MOVE on standard datasets. We fix $c = 1$ and $\gamma = 10^6$. The best performing cost ratio for each setting is bolded.

| Task | Dataset | $T$ | LOC | | ExactGP | | MOVE | |
|---|---|---|---|---|---|---|---|---|
| | | | Acc. ratio | Cost ratio | Acc. ratio | Cost ratio | Acc. ratio | Cost ratio |
| Class. | CIFAR-10 | 3 | 0.04 | 0.0568 | 0.04 | 0.0640 | 0.01 | **0.0433** |
| | | 5 | 0.03 | 0.0484 | 0.03 | 0.0563 | 0.02 | **0.0423** |
| | CIFAR-100 | 3 | 0.19 | 0.0633 | 0.19 | 0.0638 | 0.07 | **0.0404** |
| | | 5 | 0.16 | 0.0525 | 0.20 | 0.0592 | 0.07 | **0.0334** |
| | ImageNet | 3 | -0.03 | 0.0016 | 0.02 | 0.0144 | 0.02 | **0.0117** |
| | | 5 | -0.04 | 0.0034 | 0.02 | 0.0107 | 0.01 | **0.0099** |
| Seg. | VOC-Seg | 3 | 0.06 | 0.1516 | 0.08 | 0.1647 | 0.00 | **0.1045** |
| | | 5 | 0.05 | 0.1308 | 0.07 | 0.1465 | 0.01 | **0.1041** |
| Det. | VOC-Det | 3 | 0.54 | 0.1538 | 0.18 | **0.0587** | 0.19 | 0.0633 |
| | | 5 | 0.40 | 0.1159 | 0.26 | 0.0785 | 0.21 | **0.0614** |

## 4.1. Data and Methods

We evaluate MOVE in multi-source data collection scenarios based on standard vision benchmarks. For image classification, we use CIFAR-10, CIFAR-100 (Krizhevsky et al., 2009), and ImageNet (Deng et al., 2009) and train a ResNet18 (He et al., 2016) model to meet a target top-1 accuracy. For semantic segmentation, we use the PASCAL VOC 2012 with SBD augmentations (Hariharan et al., 2011; Everingham et al., 2015) and train DeepLabV3 (Chen et al., 2017), evaluating mean intersection-over-union (mIoU) against a target level. For object detection, we use PASCAL VOC with an SSD300 detector (Liu et al., 2016) and measure performance using mean average precision (mAP).

In all cases, we compare MOVE with two baselines. LOC (Mahmood et al., 2022) fits additive power law regression models to multi-source learning curves and estimates the distribution of the minimum-cost data requirement via bootstrap and density estimation. ExactGP replaces our monotonic variational GP with an exact GP that models the learning curve without monotonicity constraints, while sharing the same expected shortfall objective, thereby isolating the effect of our monotonic variational formulation on data collection performance.

In our experiments, we initialize with an initial labeled set of size $\mathbf{n}_0$. We set $\mathbf{n}_0 = 10\%$ of the full dataset for classification and segmentation, and $\mathbf{n}_0 = 20\%$ for detection. Each dataset is partitioned into $K = 5$ sources. At each round, the data collection method solves for the amount of data to acquire from each source, after which we train the downstream task model on the newly collected dataset and evaluate the achieved performance. All methods share the same downstream training pipelines and evaluation protocols. This setup reflects a practical multi-source labeling scenario with a limited initial budget. Further implementation details are in Appendix B.

## 4.2. Main Results

We consider $T \in \{3, 5\}$ rounds and evaluate all methods using two metrics. First, the accuracy ratio measures how close the final performance is to the target score, defined as $V(\mathbf{n}_T)/V^* - 1$, where values closer to zero indicate closer attainment of the target. Second, the cost ratio measures the normalized data collection cost, defined as $\mathbf{c}^\top(\mathbf{n}_T - \mathbf{n}_0)/\mathbf{c}^\top(\mathbf{n}_{\max} - \mathbf{n}_0)$. Here, $\mathbf{n}_{\max}$ denotes the full dataset. Together, these metrics evaluate both efficiency and reliability. The cost ratio quantifies acquisition cost, while the accuracy ratio indicates whether the final model meets the target. Our objective is to achieve the target score $V^*$ while minimizing additional data collection cost.

**Results on Benchmark Datasets.** Table 1 aggregates the accuracy ratios and cost ratios for each setting. Across most settings, MOVE achieves accuracy ratios closer to zero than both LOC and ExactGP, indicating that it reaches the target score more precisely while avoiding unnecessary over-collection. Overall, across classification (CIFAR-10, CIFAR-100), segmentation, and detection benchmarks with $T \in \{3, 5\}$, MOVE lowers the cost ratio by approximately 33% compared to LOC. On ImageNet, LOC fails to reach the target score, as indicated by its negative accuracy ratio. In contrast, MOVE consistently meets the target across all datasets, demonstrating the advantage of a flexible non-parametric learning curve model. ExactGP underperforms MOVE in terms of cost ratio across most settings, with the exception of VOC-Det at $T = 3$, demonstrating that enforcing coordinate-wise monotonicity is critical for reliable learning curve extrapolation and cost-efficient data collection. As a result, MOVE produces target-aware, cost-efficient data collection decisions across diverse tasks.

Figure 1 compares MOVE with LOC on CIFAR-10 and VOC detection while sweeping the target score $V^*$, with the planning horizon fixed to $T = 5$. Across the target scores,

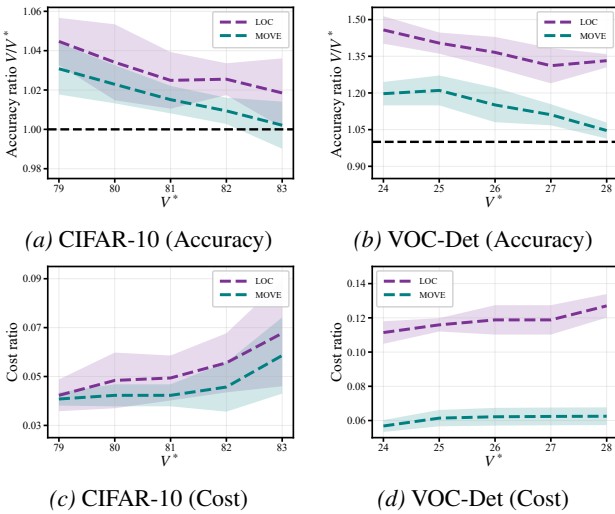

*(a)* CIFAR-10 (Accuracy)          *(b)* VOC-Det (Accuracy)

*(c)* CIFAR-10 (Cost)          *(d)* VOC-Det (Cost)

*Figure 1.* Mean $\pm$ standard deviation over 10 seeds across different targets $V^*$ with $T = 5$ fixed. We report accuracy, $V(\mathbf{n}_T)/V^*$, and the cost ratio. In the accuracy plot, a value of 1 means the target is met exactly.

MOVE stays closer to the black target line and consistently achieves lower cost ratios than LOC. This indicates that MOVE meets each target score more tightly while requiring less additional data. The cost gap is especially pronounced for VOC detection, where Figure 1d indicates that MOVE achieves an average cost-ratio reduction of about $48\%$ relative to LOC. Overall, these results suggest that MOVE is robust to the choice of target score.

**Empirical Validation of the Penalty Term**  We empirically validate Theorem 3.4 and Theorem 3.5 by varying only the terminal penalty. We use the same CIFAR-10 experimental setup as in the main results and keep the monotonic variational GP learning curve predictor fixed. To focus on the effect of the terminal penalty, we set the planning horizon to $T = 1$. We then evaluate two terminal penalties: failure probability and expected shortfall.

Figure 2 illustrates the optimization dynamics induced by the two terminal penalties over the first 50 optimization steps with $T = 1$. It reports the terminal penalty-gradient norm in Figure 2b and the corresponding success probability trajectory $F(\mathbf{n})$ in Figure 2a. Consistent with Theorem 3.4, Figure 2b shows that the failure-probability penalty produces a near-zero terminal gradient even when performance remains below the target, leaving the optimization landscape nearly flat in the under-target regime. In contrast, Figure 2a further shows that the expected-shortfall penalty continues to increase the success probability until the target region is approached.

**Learning Curve Model Ablation**  We evaluate the learning curve model in an extrapolation setting because the data

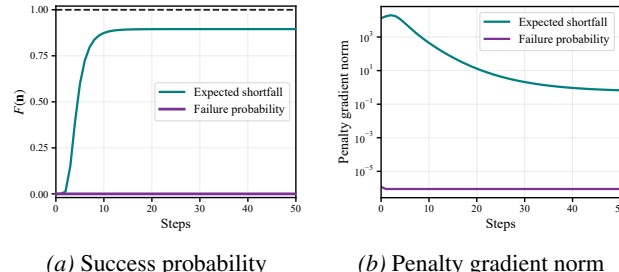

*(a)* Success probability          *(b)* Penalty gradient norm

*Figure 2.* Comparisons of the expected-shortfall and failure-probability penalties on CIFAR-10 over the first 50 optimization steps, showing step-wise trajectories of the penalty gradient norm and the predicted success model value $F(\mathbf{n})$.

collection decision must predict performance at dataset sizes beyond those observed in the current training runs. Using real training pipelines, obtaining dense ground-truth performance measurements in these extrapolation regions would require many additional trainings across a wide range of multi-source dataset compositions, which is prohibitively expensive. To enable controlled comparisons with known ground truth, we define a synthetic coordinate-wise monotone learning curve using a five-dimensional additive power law, $f^\star(\mathbf{n}) = \sum_{k=1}^{5} a_k n_k^{b_k} + c$, which captures the standard power-law behavior of learning curves; we calibrate its parameters using learning curve observations obtained from our CIFAR-10 training pipeline. We train on a lower-data region and test extrapolation on a higher-data region, comparing MOVE with DSVGP (Padidar et al., 2021) and SVGP (Hensman et al., 2015) using MSE and NLL under $\mathcal{N}(\mu(\mathbf{n}), \sigma^2(\mathbf{n}))$. We set monotonicity weight, $\lambda = 0.001$ to demonstrate that even a minimal enforcement of monotonicity is sufficient to yield consistent improvements over unconstrained baselines.

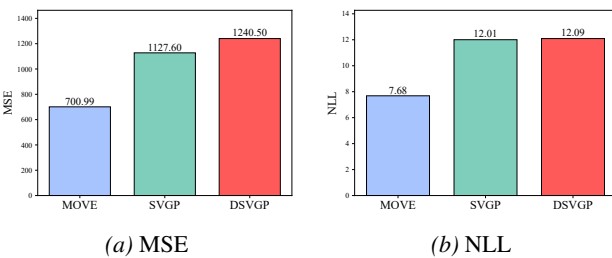

*(a)* MSE          *(b)* NLL

*Figure 3.* Predictive performance comparison in an extrapolation regime. The left subplot shows the mean squared error (MSE) and the right subplot shows the negative log-likelihood (NLL) for MOVE and variational GP baselines.

Figure 3 reports extrapolation performance in terms of MSE and NLL for MOVE, SVGP, and DSVGP. MOVE achieves the lowest MSE and NLL, indicating more accurate predictions and more reliable predictive uncertainty when generalizing from the lower-data region to the unobserved higher-data region. These results suggest that enforcing

coordinate-wise monotonicity helps avoid non-monotonic behavior when extrapolating beyond the observed range, which can degrade predictive fit for unconstrained variational GPs. Such accurate extrapolation directly translates to better data collection decisions.

## 5. Conclusion and Future Work

In this paper, we studied target-driven data collection across multiple sources, where the goal is to decide how much additional data to acquire from each source in order to reach a target performance at minimal cost. We highlighted two key challenges. The first is learning a reliable multi-source learning curve without assuming a specific parametric form. The second is designing an objective that provides informative optimization signals when performance is far below the target. To address these challenges, we proposed MOVE, which models performance as a coordinate-wise nondecreasing function of per-source sample sizes via a monotonic variational Gaussian process with virtual derivative observations and decides data collection using an expected shortfall terminal objective. Across classification, segmentation, and detection benchmarks, MOVE consistently reaches the target more tightly while reducing data acquisition cost compared to both baselines. Our analysis further explains why the expected shortfall objective avoids the flat-gradient behavior of the failure probability penalty, enabling reliable optimization even when the downstream model performance is below the target.

We discuss two promising directions for future work that extend MOVE beyond its current scope. MOVE currently determines source-wise acquisition sizes. Afterward, samples are selected uniformly at random within each source. Coupling MOVE with active learning or within-source selection would allow the framework to prioritize more informative examples after the source-level decision is made. This has the potential to improve sample efficiency and further reduce the cost required to reach the target. In addition, while we evaluate MOVE on standard vision benchmarks, broader validation on larger-scale benchmarks and real-world datasets remains important. Such studies would provide a stronger assessment of robustness under diverse data distributions.

## Acknowledgements

This research was supported in part by the Korea Planning & Evaluation Institute of Industrial Technology (KEIT) funded by the Ministry of Trade, Industry and Resources (No. RS-2025-25458052, Development of Core Technologies for Manufacturing Foundation Models / No. RS-2025-25450032, Development of AI-Driven Non-Destructive Inspection Equipment for Defect Detection and Analysis of High-Bandwidth Memory); in part by the Institute of Information & Communications Technology Planning & Evaluation (IITP)-Global Data-X Leader HRD program grant funded by the Korea government (MSIT) (IITP-2026-RS-2024-00441244); in part by the Korea Institute for Advancement of Technology (KIAT) grant funded by the Korea Government (MOTIE) (RS-2024-00409092, 2024 HRD Program for Industrial Innovation); in part by the National Research Foundation of Korea (NRF) grant funded by the Korea government (MSIT) (RS-2025-16072964).

## Impact Statement

This paper proposes MOVE, a method for deciding multi-source data collection under labeling and compute budgets. MOVE aims to improve data collection efficiency by deciding how much data to acquire from each source to reach a target performance while avoiding unnecessary annotation and training. By reducing redundant labeling and computation, this line of work can lower operational costs and energy use.

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

# A. Proofs

## A.1. Monotonic Variational Gaussian Process ELBO derivation

We use the notation from 3.1. Since differentiation is linear, derivatives of a GP are jointly Gaussian with function values (Solak et al., 2002). Consequently, $(\mathbf{f}, \mathbf{g})$ is jointly Gaussian under the prior, with block covariance matrix $K_{\text{joint}}$ defined below. Monotonicity is incorporated separately through the virtual probit likelihood on $\mathbf{g}$.

$$\begin{pmatrix} \mathbf{f} \\ \mathbf{g} \end{pmatrix} \sim \mathcal{N}\left( \begin{pmatrix} \boldsymbol{\mu}_{\mathbf{f}} \\ \boldsymbol{\mu}_{\mathbf{g}} \end{pmatrix}, K_{\text{joint}} \right), \qquad K_{\text{joint}} := \begin{pmatrix} K_{\mathbf{ff}} & K_{\mathbf{fg}} \\ K_{\mathbf{gf}} & K_{\mathbf{gg}} \end{pmatrix},$$

where $K_{\mathbf{ff}} := \text{Cov}(\mathbf{f}, \mathbf{f})$, $K_{\mathbf{fg}} := \text{Cov}(\mathbf{f}, \mathbf{g})$, $K_{\mathbf{gf}} := K_{\mathbf{fg}}^{\top}$, and $K_{\mathbf{gg}} := \text{Cov}(\mathbf{g}, \mathbf{g})$. Since the mean is constant $\mu_0$,

$$\boldsymbol{\mu}_{\mathbf{f}} = \mu_0 \mathbf{1}_N, \qquad \boldsymbol{\mu}_{\mathbf{g}} = \mathbf{0}.$$

Introduce inducing locations $Z := \{\mathbf{z}_r\}_{r=1}^{M}$ and define inducing variables consisting of function values and coordinate-wise directional derivatives:

$$\mathbf{u}_f := \big( f(\mathbf{z}_1), \ldots, f(\mathbf{z}_M) \big)^{\top}, \qquad \mathbf{u}_g := \big( e_k^{\top} \nabla f(\mathbf{z}_r) \big)_{r=1,\ldots,M;\, k=1,\ldots,K}$$

$$\mathbf{u} := \begin{pmatrix} \mathbf{u}_f \\ \mathbf{u}_g \end{pmatrix} \in \mathbb{R}^{M_u}, \qquad M_u := M(1 + K).$$

Under the GP prior, $\mathbf{u}$ is Gaussian:

$$p(\mathbf{u}) = \mathcal{N}(\boldsymbol{\mu}_{\mathbf{u}}, K_{\mathbf{uu}}), \qquad \boldsymbol{\mu}_{\mathbf{u}} = \begin{pmatrix} \mu_0 \mathbf{1}_M \\ \mathbf{0} \end{pmatrix},$$

where $K_{\mathbf{uu}}$ is the block covariance of $(\mathbf{u}_f, \mathbf{u}_g)$. Let $\mathbf{w} := \begin{pmatrix} \mathbf{f} \\ \mathbf{g} \end{pmatrix}$. Then $(\mathbf{w}, \mathbf{u})$ is jointly Gaussian under the prior, so the exact conditional is

$$p(\mathbf{w} \mid \mathbf{u}) = \mathcal{N}\big( \boldsymbol{\mu}_{\mathbf{w} \mid \mathbf{u}},\, K_{\mathbf{w} \mid \mathbf{u}} \big),$$

with

$$\boldsymbol{\mu}_{\mathbf{w} \mid \mathbf{u}} = \boldsymbol{\mu}_{\mathbf{w}} + K_{\mathbf{wu}} K_{\mathbf{uu}}^{-1} \big( \mathbf{u} - \boldsymbol{\mu}_{\mathbf{u}} \big), \qquad K_{\mathbf{w} \mid \mathbf{u}} = K_{\mathbf{ww}} - K_{\mathbf{wu}} K_{\mathbf{uu}}^{-1} K_{\mathbf{uw}}.$$

We choose a Gaussian variational distribution over inducing variables

$$q(\mathbf{u}) = \mathcal{N}(\mathbf{m}, S),$$

and define the joint variational distribution

$$q(\mathbf{w}, \mathbf{u}) = p(\mathbf{w} \mid \mathbf{u})\, q(\mathbf{u}).$$

Using the augmented model

$$p(\mathbf{y}, \mathbf{s}, \mathbf{w}, \mathbf{u}) = p(\mathbf{u})\, p(\mathbf{w} \mid \mathbf{u})\, p(\mathbf{y} \mid \mathbf{f})\, p(\mathbf{s} \mid \mathbf{g}).$$

the marginal likelihood satisfies

$$\log p(\mathbf{y}, \mathbf{s}) = \log \iint p(\mathbf{y}, \mathbf{s}, \mathbf{w}, \mathbf{u})\, d\mathbf{w}\, d\mathbf{u}.$$

Applying Jensen's inequality with respect to $q(\mathbf{w}, \mathbf{u})$ yields

$$\log p(\mathbf{y}, \mathbf{s}) \geq \iint q(\mathbf{w}, \mathbf{u}) \log \frac{p(\mathbf{y}, \mathbf{s}, \mathbf{w}, \mathbf{u})}{q(\mathbf{w}, \mathbf{u})}\, d\mathbf{w}\, d\mathbf{u} =: \text{ELBO}.$$

Substituting $q(\mathbf{w}, \mathbf{u}) = p(\mathbf{w} \mid \mathbf{u}) q(\mathbf{u})$ and cancelling the shared conditional $p(\mathbf{w} \mid \mathbf{u})$ gives

$$\text{ELBO} = \mathbb{E}_{q(\mathbf{f})}[\log p(\mathbf{y} \mid \mathbf{f})] + \mathbb{E}_{q(\mathbf{g})}[\log p(\mathbf{s} \mid \mathbf{g})] - \text{KL}\big( q(\mathbf{u}) \,\|\, p(\mathbf{u}) \big),$$

where the weighted virtual likelihood enters only through $\mathbb{E}_{q(\mathbf{g})}[\log p(\mathbf{s} \mid \mathbf{g})]$, so any multiplicative normalizing constant in $p(\mathbf{s} \mid \mathbf{g})$ that is independent of the variational parameters $(\mathbf{m}, S)$ contributes only an additive constant to the ELBO and does not affect its maximization. Using the product structure of the likelihoods,

$$\mathbb{E}_{q(\mathbf{f})}[\log p(\mathbf{y} \mid \mathbf{f})] = \sum_{i=1}^{N} \mathbb{E}_{q(f(\mathbf{x}_i))}\big[\log \mathcal{N}(y_i \mid f(\mathbf{x}_i), \sigma^2)\big],$$

$$\mathbb{E}_{q(\mathbf{g})}[\log p(\mathbf{s} \mid \mathbf{g})] = \lambda \sum_{j=1}^{J} \sum_{k=1}^{K} \mathbb{E}_{q(g_{j,k})}\big[\log \Phi(g_{j,k})\big],$$

and

$$\mathrm{KL}\big(q(\mathbf{u}) \,\|\, p(\mathbf{u})\big) = \mathrm{KL}\big(\mathcal{N}(\mathbf{m}, S) \,\|\, \mathcal{N}(\boldsymbol{\mu}_{\mathbf{u}}, K_{\mathbf{uu}})\big).$$

After maximizing ELBO over $(\mathbf{m}, S)$ and kernel hyperparameters, we obtain $q(\mathbf{u}) = \mathcal{N}(\mathbf{m}, S)$. For any $\mathbf{n} \in \mathbb{R}^K$, the predictive marginal is Gaussian,

$$q\big(f(\mathbf{n})\big) = \mathcal{N}\big(\hat{\mu}(\mathbf{n}), \hat{\sigma}^2(\mathbf{n})\big),$$

where we define

$$K_{\mathbf{nu}} := \mathrm{Cov}\big(f(\mathbf{n}), \mathbf{u}\big), \qquad K_{\mathbf{uu}} := \mathrm{Cov}(\mathbf{u}, \mathbf{u}), \qquad K_{\mathbf{nn}} := k(\mathbf{n}, \mathbf{n}).$$

Then

$$\hat{\mu}(\mathbf{n}) = \mu_0 + K_{\mathbf{nu}} K_{\mathbf{uu}}^{-1}\big(\mathbf{m} - \boldsymbol{\mu}_{\mathbf{u}}\big),$$

and

$$\hat{\sigma}^2(\mathbf{n}) = K_{\mathbf{nn}} + K_{\mathbf{nu}} K_{\mathbf{uu}}^{-1}(S - K_{\mathbf{uu}}) K_{\mathbf{uu}}^{-1} K_{\mathbf{un}}.$$

The second ELBO term depends only on one-dimensional marginals $q(g_{j,k}) = \mathcal{N}(\mu_{j,k}, \sigma_{j,k}^2)$, and we approximate $\mathbb{E}_{q(g_{j,k})}[\log \Phi(g_{j,k})]$ using Monte Carlo sampling.

## A.2. Proof of theorem 3.4

*Proof.* By definition, $R(\mathbf{n}) = \Phi\big(\alpha(\mathbf{n})\big)$ with $\alpha(\mathbf{n}) = \big(V^* - \mu(\mathbf{n})\big)/\sigma(\mathbf{n})$. Since $\Phi'(z) = \varphi(z)$, the chain rule gives

$$\nabla_{\mathbf{n}} R(\mathbf{n}) = \varphi\big(\alpha(\mathbf{n})\big) \nabla_{\mathbf{n}} \alpha(\mathbf{n}).$$

To bound $\nabla_{\mathbf{n}} \alpha(\mathbf{n})$, write

$$\alpha(\mathbf{n}) = \frac{V^* - \mu(\mathbf{n})}{\sigma(\mathbf{n})} \quad \Rightarrow \quad \nabla_{\mathbf{n}} \alpha(\mathbf{n}) = \frac{-\sigma(\mathbf{n}) \nabla_{\mathbf{n}} \mu(\mathbf{n}) - (V^* - \mu(\mathbf{n})) \nabla_{\mathbf{n}} \sigma(\mathbf{n})}{\sigma(\mathbf{n})^2}.$$

Taking norms and applying the triangle inequality yields

$$\|\nabla_{\mathbf{n}} \alpha(\mathbf{n})\| \leq \frac{\sigma(\mathbf{n}) \|\nabla_{\mathbf{n}} \mu(\mathbf{n})\| + |V^* - \mu(\mathbf{n})| \|\nabla_{\mathbf{n}} \sigma(\mathbf{n})\|}{\sigma(\mathbf{n})^2}.$$

Using $V^* - \mu(\mathbf{n}) = \alpha(\mathbf{n})\sigma(\mathbf{n})$, this becomes

$$\|\nabla_{\mathbf{n}} \alpha(\mathbf{n})\| \leq \frac{\sigma(\mathbf{n}) \|\nabla_{\mathbf{n}} \mu(\mathbf{n})\| + |\alpha(\mathbf{n})| \sigma(\mathbf{n}) \|\nabla_{\mathbf{n}} \sigma(\mathbf{n})\|}{\sigma(\mathbf{n})^2} = \frac{\|\nabla_{\mathbf{n}} \mu(\mathbf{n})\| + |\alpha(\mathbf{n})| \|\nabla_{\mathbf{n}} \sigma(\mathbf{n})\|}{\sigma(\mathbf{n})}.$$

Let $L_\mu := \sup_{\mathbf{n} \in \mathcal{F}} \|\nabla_{\mathbf{n}} \mu(\mathbf{n})\| < \infty$ and $L_\sigma := \sup_{\mathbf{n} \in \mathcal{F}} \|\nabla_{\mathbf{n}} \sigma(\mathbf{n})\| < \infty$ (which are finite under our smoothness assumptions), and note that $\sigma(\mathbf{n}) \geq \sigma_{\min} > 0$ by Assumption 3.3. Then

$$\|\nabla_{\mathbf{n}} \alpha(\mathbf{n})\| \leq \frac{L_\mu + L_\sigma |\alpha(\mathbf{n})|}{\sigma_{\min}}.$$

Combining this with the expression for $\nabla_{\mathbf{n}} R(\mathbf{n})$ gives

$$\|\nabla_{\mathbf{n}} R(\mathbf{n})\| = \varphi\big(\alpha(\mathbf{n})\big) \|\nabla_{\mathbf{n}} \alpha(\mathbf{n})\| \leq \varphi\big(\alpha(\mathbf{n})\big) \frac{L_\mu + L_\sigma |\alpha(\mathbf{n})|}{\sigma_{\min}}.$$

Setting $C_1 := L_\mu/\sigma_{\min}$ and (for $L_\mu > 0$) $C_2 := L_\sigma/L_\mu$, we obtain

$$\|\nabla_{\mathbf{n}} R(\mathbf{n})\| \le C_1 \, \varphi\big(\alpha(\mathbf{n})\big)\big(1 + C_2|\alpha(\mathbf{n})|\big).$$

Since $\varphi(\alpha) = (2\pi)^{-1/2} \exp\big(-\alpha^2/2\big)$, this bound can be written explicitly as

$$\|\nabla_{\mathbf{n}} R(\mathbf{n})\| \le \frac{C_1}{\sqrt{2\pi}} \left(1 + C_2|\alpha(\mathbf{n})|\right) \exp\big(-\alpha(\mathbf{n})^2/2\big).$$

That is, the gradient norm is bounded by a linear function of $|\alpha(\mathbf{n})|$ times a Gaussian tail. In particular, as $|\alpha(\mathbf{n})| \to \infty$, the dominant factor is the exponential term $\exp\big(-\alpha(\mathbf{n})^2/2\big)$, so $\|\nabla_{\mathbf{n}} R(\mathbf{n})\|$ decays at least exponentially fast in $|\alpha(\mathbf{n})|$. This proves the claimed exponential gradient decay.

$\square$

### A.3. Proof of theorem 3.5

*Proof.* Under the Gaussian model, the expected shortfall admits the closed form

$$S(\mathbf{n}) = (V^* - \mu(\mathbf{n})) \, \Phi\big(\alpha(\mathbf{n})\big) + \sigma(\mathbf{n}) \, \varphi\big(\alpha(\mathbf{n})\big),$$

where $\alpha(\mathbf{n}) := (V^* - \mu(\mathbf{n}))/\sigma(\mathbf{n})$ and $\Phi, \varphi$ are the standard normal CDF and PDF. Using $V^* - \mu(\mathbf{n}) = \alpha(\mathbf{n})\sigma(\mathbf{n})$ we can rewrite

$$S(\mathbf{n}) = \sigma(\mathbf{n})\big(\alpha(\mathbf{n}) \, \Phi\big(\alpha(\mathbf{n})\big) + \varphi\big(\alpha(\mathbf{n})\big)\big) = \sigma(\mathbf{n}) \, g\big(\alpha(\mathbf{n})\big)$$

with $g(a) := a \, \Phi(a) + \varphi(a)$. Using $\Phi'(a) = \varphi(a)$ and $\varphi'(a) = -a\varphi(a)$, we obtain

$$g'(a) = \Phi(a).$$

Let $\sigma = \sigma(\mathbf{n})$ and $\alpha = \alpha(\mathbf{n})$. Applying the chain rule yields

$$\nabla_{\mathbf{n}} S(\mathbf{n}) = g(\alpha) \, \nabla_{\mathbf{n}} \sigma + \sigma \, g'(\alpha) \, \nabla_{\mathbf{n}} \alpha = g(\alpha) \, \nabla_{\mathbf{n}} \sigma + \sigma \, \Phi(\alpha) \, \nabla_{\mathbf{n}} \alpha.$$

Differentiating $\alpha = (V^* - \mu)/\sigma$ gives

$$\nabla_{\mathbf{n}} \alpha = -\frac{1}{\sigma} \nabla_{\mathbf{n}} \mu - \frac{\alpha}{\sigma} \nabla_{\mathbf{n}} \sigma.$$

Substituting this expression yields

$$\nabla_{\mathbf{n}} S(\mathbf{n}) = g(\alpha) \, \nabla_{\mathbf{n}} \sigma + \sigma \, \Phi(\alpha)\left(-\frac{1}{\sigma} \nabla_{\mathbf{n}} \mu - \frac{\alpha}{\sigma} \nabla_{\mathbf{n}} \sigma\right) = -\Phi(\alpha) \, \nabla_{\mathbf{n}} \mu + \big(g(\alpha) - \alpha\Phi(\alpha)\big) \nabla_{\mathbf{n}} \sigma.$$

By the definition of $g$, $g(\alpha) - \alpha\Phi(\alpha) = \varphi(\alpha)$, and hence

$$\nabla_{\mathbf{n}} S(\mathbf{n}) = -\Phi\big(\alpha(\mathbf{n})\big) \, \nabla_{\mathbf{n}} \mu(\mathbf{n}) + \varphi\big(\alpha(\mathbf{n})\big) \, \nabla_{\mathbf{n}} \sigma(\mathbf{n}).$$

Finally, as $\alpha(\mathbf{n}) \to \infty$, we have $\Phi(\alpha(\mathbf{n})) \to 1$ and $\varphi(\alpha(\mathbf{n})) \to 0$, and thus

$$\nabla_{\mathbf{n}} S(\mathbf{n}) = -\Phi\big(\alpha(\mathbf{n})\big) \, \nabla_{\mathbf{n}} \mu(\mathbf{n}) + \varphi\big(\alpha(\mathbf{n})\big) \, \nabla_{\mathbf{n}} \sigma(\mathbf{n}) \; approaches \; - \nabla_{\mathbf{n}} \mu(\mathbf{n}).$$

This completes the proof. $\square$

### A.4. Proof of proposition 3.6

*Proof.* **Step 1: differentiability and boundedness of $\nabla R$ and $\nabla S$.** By Assumption 3.2, the predictive mean $\mu(\mathbf{n})$ and standard deviation $\sigma(\mathbf{n})$ are $C^2$ on $\mathcal{F}$ and all first and second-order derivatives are bounded. By Assumption 3.3, we have $\sigma(\mathbf{n}) \ge \sigma_{\min} > 0$ for all $\mathbf{n} \in \mathcal{F}$. Hence the standardized margin

$$\alpha(\mathbf{n}) := \frac{V^* - \mu(\mathbf{n})}{\sigma(\mathbf{n})}$$

is $C^1$ with gradient

$$\nabla_{\mathbf{n}}\alpha(\mathbf{n}) = -\frac{1}{\sigma(\mathbf{n})}\nabla_{\mathbf{n}}\mu(\mathbf{n}) - \frac{V^* - \mu(\mathbf{n})}{\sigma(\mathbf{n})^2}\nabla_{\mathbf{n}}\sigma(\mathbf{n}).$$

The normal CDF $\Phi$ and PDF $\varphi$ are smooth with bounded derivatives, so

$$R(\mathbf{n}) = \Phi\big(\alpha(\mathbf{n})\big), \qquad S(\mathbf{n}) = (V^* - \mu(\mathbf{n}))\,\Phi\big(\alpha(\mathbf{n})\big) + \sigma(\mathbf{n})\,\varphi\big(\alpha(\mathbf{n})\big)$$

are $C^1$ functions of $\mathbf{n}$. Differentiating $R$ gives

$$\nabla_{\mathbf{n}}R(\mathbf{n}) = \varphi\big(\alpha(\mathbf{n})\big)\,\nabla_{\mathbf{n}}\alpha(\mathbf{n}),$$

and $\nabla_{\mathbf{n}}S(\mathbf{n})$ is obtained by applying the product and chain rules term by term. In all cases, $\nabla_{\mathbf{n}}R$ and $\nabla_{\mathbf{n}}S$ are continuous compositions of bounded functions on the compact set $\mathcal{F}$, hence they are bounded on $\mathcal{F}$.

**Step 2: Lipschitz continuity of $\nabla R$ and $\nabla S$ on $\mathcal{F}$.** Assumption 3.2 guarantees that all partial derivatives of $\mu$ and $\sigma$ up to order two are bounded. Since $\Phi$ and $\varphi$ are smooth with bounded derivatives, it follows that all first derivatives of the components of $\nabla_{\mathbf{n}}R$ and $\nabla_{\mathbf{n}}S$ are bounded on $\mathcal{F}$. On a compact set, bounded derivatives imply Lipschitz continuity; hence there exist constants $C_R, C_S > 0$ such that

$$\|\nabla_{\mathbf{n}}R(\mathbf{n}_1) - \nabla_{\mathbf{n}}R(\mathbf{n}_2)\| \le C_R\|\mathbf{n}_1 - \mathbf{n}_2\|, \qquad \|\nabla_{\mathbf{n}}S(\mathbf{n}_1) - \nabla_{\mathbf{n}}S(\mathbf{n}_2)\| \le C_S\|\mathbf{n}_1 - \mathbf{n}_2\|$$

for all $\mathbf{n}_1, \mathbf{n}_2 \in \mathcal{F}$.

**Step 3: Lipschitz continuity of $\nabla\mathcal{L}$ on $\mathcal{F}$.** Write $\mathbf{n} = (\mathbf{n}_1, \ldots, \mathbf{n}_T)^\top$ for the stacked decision vector and view $\mathcal{L}$ in (29) as a function $\mathcal{L} : \mathcal{F} \to \mathbb{R}$ of $\mathbf{n}$. Each state $\mathbf{n}_t$ is obtained from $\mathbf{n}$ by a linear map. Hence there exists a constant $L_t < \infty$ such that

$$\|\mathbf{n}_t(\mathbf{n}_1) - \mathbf{n}_t(\mathbf{n}_2)\| \le L_t\,\|\mathbf{n}_1 - \mathbf{n}_2\| \qquad \forall t,$$

and letting $L := \max_{t \le T} L_t$ gives the uniform bound

$$\|\mathbf{n}_t(\mathbf{n}_1) - \mathbf{n}_t(\mathbf{n}_2)\| \le L\,\|\mathbf{n}_1 - \mathbf{n}_2\| \qquad \forall t.$$

The objective (29) is a finite sum of terms of the form

$$c_t^\top(\mathbf{n}_t - \mathbf{n}_{t-1})\,R(\mathbf{n}_{t-1}) \quad \text{and} \quad \gamma\,S(\mathbf{n}_T),$$

that is, compositions of the smooth functions $R$ and $S$ with linear maps of $\mathbf{n}$, and linear cost terms. Differentiating term by term and using the chain rule yields

$$\nabla_{\mathbf{n}}\mathcal{L}(\mathbf{n}) = \sum_{t=1}^{T} A_t(\mathbf{n})\,\nabla_{\mathbf{n}}R(\mathbf{n}_{t-1}) \;+\; B(\mathbf{n})\,\nabla_{\mathbf{n}}S(\mathbf{n}_T) \;+\; b,$$

for suitable matrices $A_t(\mathbf{n})$, $B(\mathbf{n})$ and constant vector $b$ whose entries remain bounded on $\mathcal{F}$. Moreover, since $\mathcal{F}$ is compact and $R, S$ are $C^1$ on $\mathcal{F}$ from Step 1, the functions $A_t(\mathbf{n})$ and $B(\mathbf{n})$ are continuous and thus bounded on $\mathcal{F}$; further, their dependence on $\mathbf{n}$ is through linear maps and the bounded functions $R$ and $S$, so $A_t$ and $B$ are Lipschitz on $\mathcal{F}$.

Using the Lipschitz bounds for $\nabla_{\mathbf{n}}R$ and $\nabla_{\mathbf{n}}S$ from Step 2, together with the linearity of the maps $\mathbf{n} \mapsto \mathbf{n}_t(\mathbf{n})$ and the above properties of $A_t(\mathbf{n})$ and $B(\mathbf{n})$, we obtain

$$\|\nabla_{\mathbf{n}}\mathcal{L}(\mathbf{n}_1) - \nabla_{\mathbf{n}}\mathcal{L}(\mathbf{n}_2)\| \le C\,\|\mathbf{n}_1 - \mathbf{n}_2\| \qquad \forall\,\mathbf{n}_1, \mathbf{n}_2 \in \mathcal{F}$$

for some constant $C > 0$ depending only on $C_R, C_S$, the cost coefficients, the time horizon $T$, and $L$. Since $R$ and $S$ are $C^1$ and the remaining terms in (29) are affine, $\mathcal{L}$ is continuously differentiable on $\mathcal{F}$.

$\square$

# B. Experiment Setup

In this section, we summarize the experimental setup used in our study. Appendix B.1 describes the data collection protocol used to construct the initial labeled set and Appendix B.2 details the model and optimization hyperparameters. All experiments are implemented in PyTorch and run on NVIDIA GeForce RTX 4090 and PRO 6000 GPU.

*Table 2.* Datasets, tasks, and evaluation metrics used in our experiments.

| Dataset | Task | Metric | Full data set size |
|---|---|---|---|
| CIFAR-10 (Krizhevsky et al., 2009) | Classification | Top-1 Accuracy | 50,000 |
| CIFAR-100 (Krizhevsky et al., 2009) | Classification | Top-1 Accuracy | 50,000 |
| ImageNet (Deng et al., 2009) | Classification | Top-1 Accuracy | 1,281,167 |
| VOC-Seg (Hariharan et al., 2011; Everingham et al., 2015) | Semantic Segmentation | Mean IoU | 10,582 |
| VOC-Det (Everingham et al., 2015) | Object Detection | Mean AP | 16,551 |

## B.1. Data collection

We summarize the data collection procedure used in our multi-source experiments. Each dataset is partitioned into $K = 5$ sources, and a data collection decision is represented as a labeled-count vector $\mathbf{n} = (n^1, \ldots, n^5)^\top$, where $n^k$ is the number of labeled data acquired from source $k$. We use these labeled-count configurations to construct training data for the learning curve models used by LOC and MOVE.

For each source $k$, we set the initial labeled size to $n_0^k = 10\%$ of the source pool (we use 20% for VOC detection). To obtain learning curve observations, we additionally construct smaller labeled sets by collecting $50\%$, $75\%$, and $100\%$ of the initial labeled set within each source, and evaluate all combinations of these per-source subsample levels across the five sources.

**Image classification.** For CIFAR-10 ,CIFAR-100 (Krizhevsky et al., 2009) and ImageNet (Deng et al., 2009), we train a ResNet18 (He et al., 2016) model and report Top-1 accuracy. We define five sources by partitioning the label space into five disjoint class groups. All models are trained with cross-entropy loss using SGD with momentum.

**Semantic segmentation.** For the PASCAL VOC 2012 benchmark (Everingham et al., 2015), we use the widely adopted trainaug split, which combines the VOC2012 segmentation training images with additional annotations from the Semantic Boundaries Dataset (SBD) (Hariharan et al., 2011). We evaluate all models on the VOC 2012 segmentation validation set and report mean intersection-over-union (mIoU). Unlabeled pixels are ignored when computing both the training loss and evaluation metrics. We train a DeepLabV3 (Chen et al., 2017) model for all segmentation experiments. The backbone is ResNet50 (He et al., 2016) pretrained on ImageNet (Deng et al., 2009).

We construct a $K = 5$ multi-source setting by grouping foreground semantic classes and assigning each trainaug image to exactly one source based on the set of foreground classes present in its segmentation mask. The five sources correspond to the following class groups: IndoorPets {cat, dog}, Transport {aeroplane, bicycle, boat, bus, car, motorbike, train}, Wildlife {bird, cow, horse, sheep}, IndoorObjects {TV monitor, sofa, bottle, potted plant, chair, dining table}, and HumanOnly {person only}. If classes from multiple groups co-occur in the same image, the image is assigned to a single source using a fixed priority rule; the HumanOnly source is used only when person is the sole foreground class.

During training, both input images and segmentation masks are resized to $320 \times 320$. We apply random horizontal flipping with probability $0.5$. Segmentation masks are resized using nearest-neighbor interpolation, while images are resized using bilinear interpolation.

**Object detection.** For object detection, we use the PASCAL VOC benchmark (Everingham et al., 2015) and follow the standard practice of combining the VOC2007 and VOC2012 datasets. Specifically, we train on the union of the VOC2007 trainval set and the VOC2012 trainval set, and evaluate on the VOC2007 test set. We report mean average precision (mAP) at IoU threshold 0.5 as the evaluation metric. All detection models are trained using an SSD300 (Liu et al., 2016) detector with a VGG backbone (Simonyan & Zisserman, 2015).

We construct a $K = 5$ multi-source setting for object detection using the same semantic class groups and single-assignment rule as in the semantic segmentation setup. Each image is assigned to exactly one source based on the set of object categories present in its detection annotations, ensuring disjoint source pools consistent with our per-source data collection formulation.

## B.2. Implementation Details

We initialize with 10% of the full training set for classification and segmentation, and with 20% for detection. We consider decision horizons of $T \in \{3, 5\}$ rounds. We summarize key hyperparameters in Table 3.

*Table 3.* Key hyperparameters used in MOVE.

| Parameter | Setting |
|---|---|
| **Monotonic GP parameter** | |
| Inducing points $M$ | 128 |
| Virtual-derivative bins $J_0$ | 5 |
| Minibatch size | 100 |
| Monotonicity weight $\lambda$ | 0.07776 |
| GP training epochs | 200 |
| **Optimization parameter** | |
| Optimizer | Adam($\beta_1 = 0.9$, $\beta_2 = 0.999$, $\epsilon = 10^{-8}$) |
| Stopping tolerance | $10^{-3}$ |
| **Data collection parameter** | |
| Optimizer | SGD(momentum $= 0.9$, weight decay $= 5 \cdot 10^{-4}$) |
| | AdamW(weight decay $= 10^{-4}$) |

We train the monotonic variational GP by maximizing the ELBO in (22), using inducing variables for both function values and coordinate-wise directional derivatives, and virtual derivative constraints implemented via the probit construction with binary variables $s_{j,k}$. For the GP prior, we use a constant mean function and a RBF covariance kernel. To incorporate derivative inducing variables and virtual derivative information, we adopt the corresponding directional derivative extension of the RBF kernel. In our multi-source setting, $\mathbf{n} \in \mathbb{N}^K$ is a $K$-dimensional vector of labeled data counts (with $K = 5$ in our main experiments). We construct a training set with multi-source design points, and use the smallest per-source count in this set as the minimum training data for placing virtual-derivative locations. For each coordinate $k \in \{1, \ldots, K\}$, we place $J_0 = 5$ candidate values by uniformly spacing points between this minimum and the per-source cap, yielding $J = J_0^K$ total virtual-derivative locations in $\mathbb{R}^K$. At these locations we fix $s_{j,k} = 1$ to encourage $\partial f(\mathbf{n})/\partial n_k \geq 0$ through the probit factors. We estimate the probit expectation terms via Monte Carlo sampling with 15 samples.

We control the influence of the virtual derivative constraints using the monotonicity weight $\lambda$ via the weighted virtual likelihood in (12). In the ELBO (22), this is equivalent to scaling the sum of virtual-derivative log-likelihood terms by $\lambda$. We set $\lambda$ proportional to the ratio $N/J$, where $J$ is the number of virtual locations and $N$ is the number of observed training runs, and fix it across experiments.

We optimize the variational parameters and kernel hyperparameters with Adam, using minibatches over the observed runs. Each run starts from the same set of initial learning curve observations and proceeds in a receding-horizon manner. At round $t$, we fit or update the monotonic GP using all collected pairs $\{(\mathbf{n}, V(\mathbf{n}))\}$. We then solve an $H = T - t + 1$ step planning problem with gradient-based optimization. In the planner, we parameterize nonnegative increments using a softplus transformation and accumulate them to obtain a coordinate-wise nondecreasing plan, enforce the per-source caps by box projection.

## C. Additional Numerical Results

This section contains expanded numerical experiments. Our key results include:

- In Appendix C.1, we evaluate the robustness of MOVE to variations in the cost parameters. We report how the performance changes as we vary the per-source cost, and show that MOVE remains stable across various settings.

- In Appendix C.2, we extend our evaluation to the $K = 2$ setting instead of the default $K = 5$. We study how MOVE behaves when the number of sources is reduced and confirm that its performance trends remain consistent.

### C.1. Robustness to the Cost Parameters

Table 4 studies sensitivity to the source-wise acquisition costs by varying the cost vector on CIFAR-10 and VOC-Det with planning horizon $T = 5$. We evaluate four cost regimes that change the relative prices across sources and report the resulting accuracy and cost ratios for LOC and MOVE. For nearly all settings for cost vector, MOVE consistently achieves lower cost ratios than LOC on both tasks while maintaining comparable accuracy ratios. This suggests that the effectiveness of MOVE is not tied to a particular cost specification, but persists across diverse cost structures.

*Table 4.* Cost sensitivity on CIFAR-10 and VOC-Det. We fix $P = 10^6$. The best performing cost ratio for each setting is bolded.

| Task | Dataset | $T$ | Cost | LOC | | MOVE | |
|---|---|---|---|---|---|---|---|
| | | | | Acc. ratio | Cost ratio | Acc. ratio | Cost ratio |
| Class. | CIFAR-10 | 5 | $(1, 1, 1, 1, 1)$ | 0.03 | 0.0484 | 0.02 | **0.0423** |
| | | | $(1, 1, 0.5, 0.5, 1)$ | 0.03 | 0.0469 | 0.02 | **0.0389** |
| | | | $(1, 0.5, 0.5, 0.1, 1)$ | 0.03 | **0.0376** | 0.03 | 0.0425 |
| | | | $(1, 0.5, 0.1, 0.1, 1)$ | 0.02 | 0.0323 | 0.03 | **0.0270** |
| Det. | VOC-Det | 5 | $(1, 1, 1, 1, 1)$ | 0.40 | 0.1159 | 0.21 | **0.0614** |
| | | | $(1, 1, 0.5, 0.5, 1)$ | 0.41 | 0.1177 | 0.17 | **0.0657** |
| | | | $(1, 0.5, 0.5, 0.1, 1)$ | 0.38 | 0.1069 | 0.21 | **0.0553** |
| | | | $(1, 0.5, 0.1, 0.1, 1)$ | 0.34 | 0.1064 | 0.23 | **0.0531** |

## C.2. Implementation on $K = 2$ case

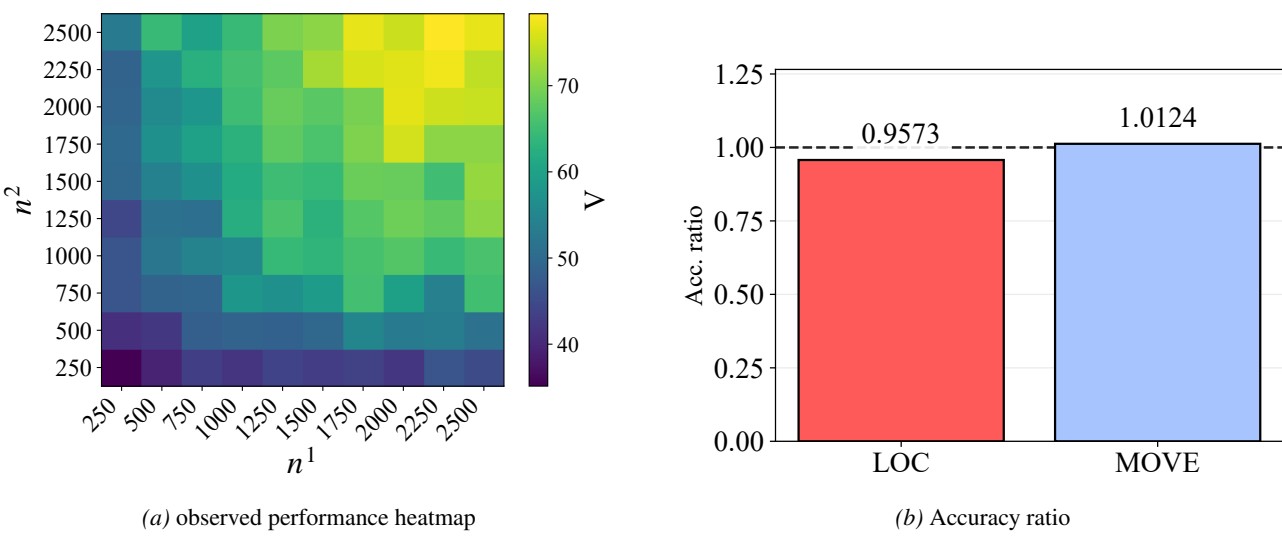

*(a)* observed performance heatmap                    *(b)* Accuracy ratio

*Figure 4.* Comparison of MOVE and LOC on CIFAR-10 with $K = 2$ over 10 random seeds. We show the observed performance heatmap over the training set and the resulting accuracy ratio.

For each source $k \in \{1, 2\}$, we set the initial labeled size to $n_0^k = 10\%$ of the source pool. To obtain learning curve observations, we then construct additional smaller labeled sets by sub-sampling from this initial set. Concretely, for each source we consider budget fractions in $\{10\%, 20\%, \dots, 100\%\}$ of $n_0^k$, which correspond to $1\%$–$10\%$ of the full source pool, and evaluate the model on combinations $(n^1, n^2)$ drawn from this grid.

Figure 4a visualizes the empirical performance surface over $(n^1, n^2)$ for the $K = 2$ CIFAR-10 (Type-2) setting, constructed from a training datasets. Figure 4b reports the resulting accuracy ratio achieved by LOC and MOVE. While LOC fails to reach the target score under this setting, MOVE consistently reaches the target with lower cost. This robustness stems from enforcing monotonic structure through virtual derivative constraints, which stabilizes extrapolation and downstream acquisition decisions in observation scarce regimes.

