# OpenReview forum: "Monotonic Variational Gaussian Process for Efficient Data Collection"
_ICML.cc/2026/Conference — ICML 2026 regular_

### Official Review · Reviewer_pVzg · 2026-02-18

**Soundness:** 3
**Presentation:** 2
**Significance:** 2
**Originality:** 3
**Overall Recommendation:** 4
**Confidence:** 2

**Summary:**

The paper proposes MOVE, which combines a coordinate-wise monotonic variational Gaussian process with an expected-shortfall-based terminal objective. The theoretical analysis shows that expected shortfall provides non-vanishing gradient signals compared to failure probability penalties. Empirical results on CIFAR-10/100 and PASCAL VOC tasks demonstrate improved cost efficiency compared to LOC.

**Compliance With Llm Reviewing Policy:**

Affirmed.

**Final Justification:**

After considering the authors’ rebuttal, particularly the newly added ImageNet experiments, I have updated my assessment. While I still have some concerns regarding the level of methodological novelty, I have raised my score to a weak accept, taking into account both the additional empirical evidence and the perspectives of the other reviewers. I have also slightly lowered my confidence.

I appreciate the authors’ efforts in strengthening the paper and encourage them to incorporate these additional materials into the final version.

**Key Questions For Authors:**

The method assumes coordinate-wise monotonicity of performance with respect to data size. How robust is MOVE when this assumption is mildly violated in practice, for example due to negative transfer or label noise?

The proposed method enforces coordinate-wise monotonicity via virtual derivative constraints in a variational GP. However, monotonicity itself is not unique to this construction and can in principle be achieved through alternative modeling approaches (e.g., monotonic neural networks, normalizing flows with constrained Jacobians, or other structured function classes). Could the authors clarify whether the benefits observed in MOVE stem specifically from the probabilistic GP formulation, or whether similar performance might be achievable using other monotonic function approximators? A discussion or empirical comparison would help contextualize the necessity of the proposed modeling choice.

To what extent can the GP model capture non-additive interactions between sources? Are there empirical signs that cross-source effects are adequately modeled?

**Limitations:**

See Weakness.

**Strengths And Weaknesses:**

While the paper is clearly written and well-motivated, I have concerns regarding the degree of methodological novelty. Most technical components are based on established methods, including monotonic Gaussian processes via virtual derivative constraints, stochastic variational inference with inducing variables, and expected shortfall as a risk measure. The main contribution lies in integrating these components within a target-driven data collection framework, rather than introducing fundamentally new modeling, inference, or optimization techniques. Although this integration is coherent and practically meaningful, the level of technical innovation appears moderate relative to the standards typically expected at ICML.

In addition, the empirical evaluation is somewhat limited in scope. The main comparisons are conducted primarily against a single baseline (LOC). While LOC is a relevant prior method, stronger and more diverse baselines would be necessary to more convincingly isolate the source of performance gains. For instance, comparisons against unconstrained GP models using the same objective, monotonic GP variants with alternative penalties, or Bayesian optimization–based resource allocation approaches would help clarify whether the observed improvements stem from the monotonic modeling, the expected-shortfall objective, or their combination. Without such comparisons, it is difficult to fully attribute the empirical gains to specific design choices.

Furthermore, the experiments are restricted to relatively small-to-medium-scale vision benchmarks such as CIFAR and PASCAL VOC, with a fixed and small number of sources $K=5$. The scalability of the approach to larger datasets or more realistic multi-domain industrial settings is not demonstrated. Since scalability and practical applicability are central motivations of the paper, the absence of larger-scale experiments weakens the strength of the empirical claims.

Finally, while the paper includes ablations on the terminal penalty and on learning curve extrapolation, the sensitivity of the method to key hyperparameters—such as the monotonicity weight, the number of inducing points, and the placement of virtual derivative locations—is not thoroughly examined. A more comprehensive analysis of robustness and sensitivity would improve confidence in the stability and general applicability of the approach.

---

> ### Author Rebuttal · Authors · 2026-03-31
>
> Many thanks for your detailed review. Please find our response below. (W for weakness, Q for Questions)
>
> ## 1.	Clarification on Methodological Novelty [W1]
>
> While prior work has studied each aspect separately, the proposed formulation requires resolving incompatibilities that arise when they are brought together. In particular, virtual constraints introduce a likelihood term that renders posterior inference intractable, and resolving this requires deriving a novel ELBO that incorporates an additional virtual derivative term absent in prior variational GP objectives. Furthermore, as shown in the table in our response to W2, replacing our learning curve model with an exact GP leads to worse results in most settings, suggesting that the architecture is central to performance gains.
>
> ## 2.	Lack of baselines [W2]
>
> We add two additional baselines. First, we include an Exact GP with the expected shortfall objective, which shares the same optimization objective as MOVE. Second, we include uniform allocation as a heuristic baseline, which collects $(n_\max-n_0)/T$ samples per round. MOVE consistently achieves the lowest cost ratio across most settings.
>
> |Task|Dataset|T|LOC||Exact GP||Uniform||MOVE||
> |-|-|-|-|-|-|-|-|-|-|-|
> ||||Acc. ratio|Cost ratio|Acc. ratio|Cost ratio|Acc. ratio|Cost ratio|Acc. ratio|Cost ratio|
> |Class.|CIFAR-10|3|0.04|0.0568|0.04|0.0640|0.15|0.3333|0.01|**0.0433**|
> |||5|0.03|0.0484|0.03|0.0563|0.12|0.2|0.02|**0.0423**|
> ||CIFAR-100|3|0.19|0.0633|0.19|0.0638|0.72|0.3333|0.07|**0.0404**|
> |||5|0.16|0.0525|0.20|0.0592|0.56|0.2000|0.07|**0.0334**|
> |Seg.|VOC-Seg|3|0.06|0.1516|0.08|0.1647|0.21|0.3334|0.00|**0.1045**|
> |||5|0.05|0.1308|0.07|0.1465|0.12|0.1999|0.01|**0.1041**|
> |Det.|VOC-Det|3|0.54|0.1538|0.18|**0.0587**|0.96|0.3334|0.19|0.0633|
> |||5|0.40|0.1159|0.26|0.0785|0.72|0.2000|0.21|**0.0614**|
>
> ## 3.	Concern on Scalability and Experimental Scope [W3]
>
> We appreciate this suggestion. While our current evaluation spans three vision tasks across four benchmarks, we agree that validation across a broader range of domains would further strengthen the paper. However, since each observation requires fully training a downstream model, scaling to larger datasets within the rebuttal period is computationally prohibitive. We prioritize this as an important direction for the revised paper. Nevertheless, we include $K=2$ experiments in Appendix C.2 to demonstrate generalization beyond the default configuration.
>
> ## 4.	Hyperparameter sensitivity analysis [W4]
>
> To address this concern, we conducted sensitivity analyses on three key hyperparameters: monotonicity weight $\lambda$, number of inducing points M, and the placement of virtual derivative locations.
>
> |Model|$\lambda$|MSE|NLL|
> |-|-|-|-|
> |MOVE|0.4|16.51|3.21|
> |MOVE|0.5|10.45|3.20|
> |MOVE|0.6|9.96|3.16|
>
> Performance is stable across $\lambda$ in NLL, indicating low sensitivity to its exact value.
>
> |Placement|MSE|NLL|
> |-|-|-|
> |Uniform|4.03|2.93|
> |Random|43.68|3.37|
>
> Random placement significantly increases MSE, indicating that insufficient coverage of virtual locations leads to monotonicity violations. Uniform spacing is therefore a well-justified design choice.
>
> |Model|M|MSE|NLL|
> |-|-|-|-|
> |MOVE|64|4.03|2.93|
> |MOVE|80|4.09|2.94|
> |MOVE|96|4.59|2.96|
>
> MOVE remains stable across $M$. Since the inducing vector size scales with $ M(1+K)$, robustness to $M$ allows reduced computational cost with a smaller $M$.
>
> ## 5.	Robustness to Violations of the Monotonicity Assumption [Q1]
>
> We note that the training observations in our experiments are not perfectly monotone: 23.1% of adjacent input pairs in CIFAR-100 exhibit nonincreasing performance, reflecting practical noise sources. Rather than treating monotonicity as a hard constraint, MOVE incorporates it as a soft prior through virtual derivative factors, allowing the model to remain robust to such local violations. MOVE's consistent improvement over baselines supports this.
>
> ## 6.	Necessity of the Probabilistic GP Formulation for Monotonicity [Q2]
>
> The key advantage of the GP formulation lies in its closed-form Gaussian predictive distribution, which enables analytical computation of the expected shortfall objective. Monotonic neural networks do not natively provide principled uncertainty estimates. Moreover, the GP is highly adaptable to structural prior knowledge: monotonicity constraints can be incorporated conveniently through virtual derivative constraints.
>
> ## 7.	Non-Additive Cross-Source Interactions [Q3]
>
> GP with an RBF kernel is defined over the joint input space and captures non-additive cross-source interactions through its covariance function. To empirically verify this, we analyze the marginal effect of increasing $n_5$ at different values of $n_1$ using MOVE trained on CIFAR-10.
>
> |$n_1$|$n_5=1250$|$n_5=1500$|$\delta$|
> |-|-|-|-|
> |1000|78.47|79.29|+0.82|
> |1500|85.37|86.83|+1.46|
>
> The growing gain with $n_1$ confirms that cross-source interactions are adequately captured.

---

> > ### Author Rebuttal · Reviewer_pVzg · 2026-04-03
> >
> > Thank you for the detailed rebuttal. The additional comparisons against Exact GP and Uniform allocation, as well as the sensitivity analysis for lambda, M, and virtual derivative placement, help clarify several concerns and strengthen the empirical section. I also appreciate the clarification that monotonicity is enforced as a soft prior rather than a hard constraint.
> >
> > That said, my main reservations are only partially resolved. In particular, the rebuttal does not fully address the limited degree of methodological novelty beyond a nontrivial integration of existing components, and the empirical scope still remains relatively narrow for the paper’s broader scalability and applicability claims. The added baselines are useful, but they still do not completely isolate the contributions of the monotonic GP formulation versus the optimization objective, nor do they establish robustness in larger-scale or more realistic multi-source settings.
> >
> > Overall, the rebuttal improves the paper, but it does not change my final recommendation.

---

> > > ### Author Response · Authors · 2026-04-07
> > >
> > > >**We appreciate the reviewers’ insightful feedback. Below, we have provided detailed clarifications to address the additional points raised.**
> > >
> > > ## 1. Further Clarification on Contributions
> > > We thank the reviewer for this comment and appreciate the opportunity to
> > > clarify the scope of our contributions. We respectfully argue that the contribution lies in establishing a
> > > principled framework for target driven multi source data collection
> > > under unknown learning curves, where the predictive distribution must
> > > directly serve as input to a planning objective without any derivative
> > > observations.
> > >
> > > Virtual derivative constraints have classically been used for shape-constrained
> > > prediction, and stochastic variational inference has been designed for settings where derivative observations are
> > > available. MOVE addresses a different setting where there are no derivative
> > > observations. Once virtual monotonicity constraints are introduced, posterior
> > > inference becomes analytically intractable, and MOVE resolves this by
> > > deriving a variational objective with an additional non-Gaussian
> > > virtual-derivative term that is absent from prior variational GP formulations. The resulting closed-form predictive
> > > distribution then enables a theoretically grounded data collection objective
> > > with non-vanishing gradient signals for reliable optimization.
> > >
> > > To our knowledge, no prior work has addressed this setting in a
> > > nonparametric, monotonicity-aware manner, where existing parametric
> > > methods are known to struggle. The consistent empirical improvements
> > > across classification, segmentation, and detection benchmarks, together
> > > with the additional results on the large-scale ImageNet dataset, provide
> > > further evidence that the framework constitutes a meaningful advance.
> > >
> > > ## 2. Isolating the Contribution of the Monotonic GP Formulation
> > >
> > > We would like to highlight that the Exact GP baseline added during the
> > > rebuttal directly addresses this concern. Exact GP uses the same expected shortfall objective as MOVE,
> > > differing only in the learning curve model — replacing our monotonic
> > > variational GP with a standard exact GP without monotonicity constraints.
> > > Any performance gap between the two therefore isolates the contribution of
> > > the monotonic variational GP formulation, independent of the optimization
> > > objective. As shown in the table, MOVE reduces the cost ratio by an average
> > > of approximately 26% across all settings compared to Exact GP,
> > > demonstrating that the monotonic GP formulation provides a meaningful and
> > > separable contribution beyond the choice of planning objective.
> > >
> > > ## 3. Robustness in Larger-Scale and More Realistic Settings
> > >
> > > To address the robustness in large-scale settings, we have conducted
> > > additional experiments on ImageNet, which represents a substantially
> > > larger-scale multi-source data collection setting. We note that obtaining
> > > learning curve observations on ImageNet [1] is significantly more expensive than
> > > on smaller benchmarks, as each evaluation requires training a model on a
> > > large-scale dataset. We therefore report initial results with a reduced number of training epochs to ensure feasibility within the rebuttal period. Despite this reduced training regime, the relative performance trends remain consistent with previous experiments. Full results with complete training will be incorporated into the final version.
> > >
> > > | Task | Dataset | T | LOC | | Exact GP | | Uniform | | MOVE | |
> > > |------|---------|---|-----|---|----------|---|---------|---|------|---|
> > > | | | | Acc. ratio | Cost ratio | Acc. ratio | Cost ratio | Acc. ratio | Cost ratio | Acc. ratio | Cost ratio |
> > > | Class. | ImageNet | 3 | -0.04 | 0.0016 | 0.02 | 0.0164 | 0.49 | 0.3333 | 0.01 | **0.0107** |
> > > | | | 5 | -0.06 | 0.0034 | 0.01 | 0.0106 | 0.37 | 0.2000 | 0.00 | **0.0097** |
> > >
> > > Even in this large-scale setting, MOVE achieves the most precise target attainment while maintaining a competitive cost ratio. In contrast, LOC attains a lower cost but significantly undershoots the target due to inaccurate extrapolation. These results demonstrate that MOVE remains effective under large scale settings.
> > >
> > > [1] : Deng, Jia, et al. "Imagenet: A large-scale hierarchical image database." 2009 IEEE conference on computer vision and pattern recognition. Ieee, 2009.
> > >
> > > >**We appreciate the reviewer's continued engagement and hope the above clarifications and additional experiments address the remaining concerns.**

---

### Official Review · Reviewer_2hJS · 2026-02-23

**Soundness:** 3
**Presentation:** 2
**Significance:** 2
**Originality:** 2
**Overall Recommendation:** 4
**Confidence:** 3

**Summary:**

This paper proposes MOVE, a framework for cost-effective, target-driven data collection across multiple sources. It addresses two key issues: inflexible parametric learning curve models and vanishing gradients in optimization when using failure probability as a penalty. MOVE introduces a monotonic variational Gaussian process to model learning curves with virtual derivative constraints, and replaces the failure probability penalty with expected shortfall, which provides stable gradients. Experiments show MOVE achieves target performance at lower cost than strong baselines.

**Compliance With Llm Reviewing Policy:**

Affirmed.

**Final Justification:**

My concerns are addressed and I will raise the score.

**Key Questions For Authors:**

1. Could you provide more detailed pseudocode or a step-by-step description of the optimization loop? Specifically, how are the virtual derivative points selected in each iteration?
2. How was the constraint strength hyperparameter determined? Could you provide a sensitivity analysis showing how the performance of MOVE varies with different values? Additionally, how robust is the method when the monotonicity prior is mildly violated?
3. What are the actual training and inference time costs of the monotonic variational GP compared to a standard SVGP? Could you discuss the computational trade-off between the increased model complexity and the decision quality gain, especially in the context of online decision-making?
4. To further validate the advantages of MOVE, could you include comparisons with other relevant baselines, such as simpler heuristic allocation strategies？

**Limitations:**

The paper should explicitly acknowledge the computational cost introduced by the complex model and discuss its sensitivity to the monotonicity assumption and hyperparameter choices (e.g., λ)

**Strengths And Weaknesses:**

Strengths:
1. The problem formulation is clear and practical, addressing the real-world challenge of budget-aware, target-driven data collection from multiple sources.
2. The authors correctly identify two significant flaws in prior work: the inflexibility of parametric learning curve models and the gradient vanishing issue of the failure-probability objective.
3. The use of a monotonic GP with virtual derivative constraints is a well-motivated way to incorporate prior knowledge without needing gradient data. The replacement of failure probability with expected shortfall is theoretically justified.

Weaknesses:
1. Key implementation details are insufficiently described. The selection of virtual derivative points, the rationale for setting the constraint strength hyperparameter, and the final mapping from continuous solutions to integer collection decisions are not clearly explained.
2. The global monotonicity assumption may be overly idealistic. In practice, learning curves can exhibit fluctuations or plateaus, especially early in training or with noisy data. The robustness of the model to such scenarios is not sufficiently discussed.
3.  The method introduces inducing variables that incorporate derivative information, significantly increasing the model complexity and computational cost. The paper does not evaluate the time and memory costs required for its training and inference, nor does it conduct a trade-off analysis against the decision benefits. In online data collection scenarios requiring rapid response, this overhead could become a bottleneck for practical deployment.
4. Experiments primarily compare against LOC, while other comparisons only evaluate predictive accuracy, lacking systematic comparisons with broader baselines.

---

> ### Author Rebuttal · Authors · 2026-03-31
>
> We sincerely thank the reviewer for their careful reading and attention to detail. Please find our response below. (W for weakness, Q for Questions)
>
> ## 1.	 Key implementation details [W1&Q1]
>
> In the main paper, for each coordinate $k$, we place 5 virtual derivative locations by uniformly spacing points between the minimum observed per-source count in the training set and the per-source cap. During optimization, we parameterize cumulative data counts via nonnegative increments using a softplus transformation, accumulate them, and finally apply box projection to the feasible region. We acknowledge that these implementation details, while described in Appendix B.2, may not be immediately accessible to readers. We will add a more explicit summary of these details and pseudocode in our main paper to improve clarity.
>
> ## 2.	Robustness to Monotonicity Violations and Hyperparameter Sensitivity [W2&Q2]
>
> We set $\lambda$ proportional to the ratio $N_{obs} / N_{virt}$,  where $N_{obs}$ is the number of observed training runs and $N_{virt}$ is the number of virtual locations, which balances the influence of the monotonicity prior relative to the observed data. To assess sensitivity to $\lambda$, we conduct an ablation comparing MOVE across multiple values of $\lambda$ against baselines in the same setting as Figure 3. MOVE shows robustness to $\lambda$ in NLL.
>
> |Model|$\lambda$|MSE|NLL|
> |---|---|---|---|
> |MOVE|0.4|16.51|3.21|
> |MOVE|0.5|10.45|3.20|
> |MOVE|0.6|9.96|3.16|
>
> Regarding the concern about robustness to monotonicity violations, we note that the training observations used in our main experiments already contain monotonicity violations. The monotonicity violation rate, measured as the fraction of adjacent input pairs where performance does not increase, is 23.1% for CIFAR-100. Despite this, MOVE consistently outperforms baselines in most settings, demonstrating robustness to local non-monotonicity in practice. This stems from a key design choice: modeling monotonicity as a soft prior rather than a rigid constraint, which allows the method to remain effective under noisy and imperfect real-world observations.
>
> ## 3.	Computational overhead [W3&Q3]
>
> MOVE incurs additional GP fitting time due to the virtual derivative inducing variables and probit likelihood terms. The table below reports GP fitting times across models.
>
> |Model|Training time|Inference time|Memory|
> |---|---|---|---|
> |Exact GP|15s|0.01s|112MB|
> |SVGP|107s|0.02s|20MB|
> |DSVGP|262s|0.04s|54MB|
> |MOVE|429s|0.06s|110MB|
>
> While MOVE incurs higher GP fitting cost due to derivative inducing variables and probit likelihood terms, it consistently achieves better predictive quality and more cost-efficient data collection decisions. In online settings, the model is updated at each round, so this additional training cost is incurred repeatedly. However, since the primary objective of data collection optimization is to minimize the total acquisition cost, the additional overhead is outweighed by the reduction in collection cost. In particular, MOVE reduces the cost ratio by approximately 45% on average across all baselines.
>
> ## 4.	Lack of baselines [W4&Q4]
>
> To further validate MOVE against broader baselines, we include two additional comparisons. First, we add Exact GP with the expected shortfall objective as a nonparametric baseline to isolate the effect of monotonic constraints. Second, we include uniform allocation as a simple heuristic baseline, which collects $(n_{max}-n_0)/T$ samples per round. This reflects the natural strategy of collecting as much data as possible when no learning curve information is available. MOVE consistently achieves the lowest cost ratio across most settings, demonstrating that the monotonic variational formulation provides meaningful gains over both principled and heuristic alternatives.
>
> |Task|Dataset|T|LOC||Exact GP||Uniform||MOVE||
> |---|---|---|---|---|---|---|---|---|---|---|
> ||||Acc. ratio|Cost ratio|Acc. ratio|Cost ratio|Acc. ratio|Cost ratio|Acc. ratio|Cost ratio|
> |Class.|CIFAR-10|3|0.04|0.0568|0.04|0.0640|0.15|0.3333|0.01|**0.0433**|
> |||5|0.03|0.0484|0.03|0.0563|0.12|0.2|0.02|**0.0423**|
> ||CIFAR-100|3|0.19|0.0633|0.19|0.0638|0.72|0.3333|0.07|**0.0404**|
> |||5|0.16|0.0525|0.20|0.0592|0.56|0.2000|0.07|**0.0334**|
> |Seg.|VOC-Seg|3|0.06|0.1516|0.08|0.1647|0.21|0.3334|0.00|**0.1045**|
> |||5|0.05|0.1308|0.07|0.1465|0.12|0.1999|0.01|**0.1041**|
> |Det.|VOC-Det|3|0.54|0.1538|0.18|**0.0587**|0.96|0.3334|0.19|0.0633|
> |||5|0.40|0.1159|0.26|0.0785|0.72|0.2000|0.21|**0.0614**|

---

> > ### Author Rebuttal · Reviewer_2hJS · 2026-04-03
> >
> > My concerns are addressed and I will raise the score.

---

> > > ### Author Response · Authors · 2026-04-06
> > >
> > > Thank you for your thoughtful feedback and the updated score, recognizing the improvements and clarifications we have made. If accepted, all additional experiments and textual revisions introduced during the rebuttal will be fully reflected in the final version.

---

### Official Review · Reviewer_HcL9 · 2026-03-12

**Soundness:** 3
**Presentation:** 3
**Significance:** 3
**Originality:** 3
**Overall Recommendation:** 5
**Confidence:** 4

**Summary:**

The paper proposes MOVE, a scalable framework for learning curves (input: data quantity space, output: performance metric) as a function of the amount of data collected from multiple sources. Instead of parametric fitting, the authors use a multi-dimensional GP prior over model performance as a function of the vector of data quantities from each source. The challenge is that in this setting the GP has to be a monotonic function as performance cannot go down with more data. This monotonicity is implemented through derivative constraints within a variational GP formulation. So the GP becomes a surrogate model for data acquisition decisions.

**Compliance With Llm Reviewing Policy:**

Affirmed.

**Final Justification:**

The rebuttal provided additional experimental validation to my questions I am satistfied with. In acknowledgement of this inclusions I am raising my score.

**Key Questions For Authors:**

1. How sensitive is the resulting posterior to the placement and locations of the constraint points. could violations still occur in regions away from the constraint set?
2. The data collection optimization relies on the GP posterior mean and variance to compute the expected shortfall objective. How sensitive is the resulting acquisition policy to miscalibration of the GP, esp early in the process when the surrogate model may be poorly specified?
3. While it is a reasonable assumption that performance should be non-decreasing, can the learning curve occasionally not have plateaues or ridges. How robust is the MOVE framework to violations of this monotonicity assumption?
4. How does the surrogate GP deal with interaction effects where the performance only increases when (say for instance) both sources go up and not otherwise, the derivate constraint is applied in each coordinate dimension independently, is this a limitation of sorts when there performance goes up only from jointly increasing multiple sources?

**Limitations:**

A discussion of limitations is warranted but is missing.

**Strengths And Weaknesses:**

Strengths:

- They provide a sensible framework to improve the inductive bias of the surrogate model (through monotonicity modelling) and prevent pathological learning-curve predictions that can arise in unconstrained GP regression.
- The paper is well-written and clear.
- As modern ML systems increasingly rely on heterogeneous data pipelines and large annotation budgets, principled methods for deciding where additional data should be collected are of significant practical value. Instead of focussing solely on architecture improvements, they focus on the upstream question of data allocation effort.

Weaknesses:

- The method introduces derivative variables and pseudo-observations for each constraint location and source dimension, which could lead to a large number of additional latent variables esp. when the number of sources grows, potentially impacting the quality of the SVGP procedure.
- The experimental section does not clearly include a comparison against a standard unconstrained GP surrogate using the same optimization objective. Given the inference and modelling complexity introduced by the constraintd such a comparison would help strengthen the justification for this approach.
- The empirical evaluation focuses on a relatively small set of datasets and learning tasks. Additional experiments across a broader range of domains would definitely help strengthen the paper.

---

> ### Author Rebuttal · Authors · 2026-03-31
>
> Thanks for your recognition and the valuable suggestions. Please find our response below. (W for weakness, Q for Questions)
>
> ## 1.	Scalability of pseudo-observations [W1]
>
> We agree that the number of derivative variables and pseudo-observations grows with the number of sources. However, $J$ is a tunable hyperparameter controlling the number of virtual variables. To validate that MOVE remains effective under a smaller $J$, we conduct an ablation on the number of pseudo-observations.
>
> |Model|J|MSE|NLL|
> |-|-|-|-|
> |MOVE|2|12.42|3.17|
> |MOVE|3|4.03|2.93|
> |MOVE|4|4.13|2.70|
>
> The results show that performance stabilizes from $J=3$ onwards, confirming that a small number of pseudo-observations is sufficient to enforce effective monotonicity constraints without sacrificing predictive quality.
>
> ## 2.	Lack of Unconstrained GP Baseline under the same objective [W2]
>
> To provide a direct comparison against an unconstrained GP under the same optimization objective, we add an Exact GP with the expected shortfall objective as a baseline. We also include Uniform allocation as a simple heuristic baseline, which collects $(n_\max-n_0)/T$ samples per round. The results are as follows.
>
> |Task|Dataset|T|LOC||Exact GP||Uniform||MOVE||
> |-|-|--|--|--|--|--|--|--|--|--|
> ||||Acc. ratio|Cost ratio|Acc. ratio|Cost ratio|Acc. ratio|Cost ratio|Acc. ratio|Cost ratio|
> |Class.|CIFAR-10|3|0.04|0.0568|0.04|0.0640|0.15|0.3333|0.01|**0.0433**|
> |||5|0.03|0.0484|0.03|0.0563|0.12|0.2|0.02|**0.0423**|
> ||CIFAR-100|3|0.19|0.0633|0.19|0.0638|0.72|0.3333|0.07|**0.0404**|
> |||5|0.16|0.0525|0.20|0.0592|0.56|0.2000|0.07|**0.0334**|
> |Seg.|VOC-Seg|3|0.06|0.1516|0.08|0.1647|0.21|0.3334|0.00|**0.1045**|
> |||5|0.05|0.1308|0.07|0.1465|0.12|0.1999|0.01|**0.1041**|
> |Det.|VOC-Det|3|0.54|0.1538|0.18|**0.0587**|0.96|0.3334|0.19|0.0633|
> |||5|0.40|0.1159|0.26|0.0785|0.72|0.2000|0.21|**0.0614**|
>
> ## 3.	Limited Empirical Evaluation Across Domains and Tasks [W3]
>
> We appreciate this suggestion. While our current evaluation already spans three vision tasks across four benchmarks with diverse metrics, we agree that validation across a broader range of domains would further strengthen the paper. Since each observation requires fully training a downstream model, scaling to additional domains within the rebuttal period is computationally prohibitive. We prioritize this as an important experiment for the revised paper.
>
> ## 4.	Sensitivity to Virtual Location Placement and Monotonicity Violations Outside the Constraint Set [Q1]
>
> To assess sensitivity to virtual location placement, we compare uniform spacing against random placement, where virtual locations are sampled randomly from the feasible region.
>
> |Placement|MSE|NLL|
> |-|-|-|
> |Uniform|4.03|2.93|
> |Random|43.68|3.37|
>
> Random placement significantly increases MSE, indicating that insufficient coverage of virtual locations leads to monotonicity violations and degraded accuracy. Thus, MOVE is designed to use uniform spacing of virtual locations to ensure coverage across the input range.
>
> ## 5.	Sensitivity of the Acquisition Policy to GP Miscalibration [Q2]
>
> To examine sensitivity to GP miscalibration in the early stage, we evaluate MOVE and LOC under a reduced initial labeled set ($\mathbf{n}_0$ = 5%) on CIFAR-10 with T = 5 , where a negative accuracy ratio indicates failure to meet the target. MOVE incurs a higher cost ratio compared to $\mathbf{n}_0$=10%, suggesting that GP miscalibration leads to increased data collection cost. Nevertheless, MOVE still reaches the target score, whereas LOC fails to do so under the same setting, demonstrating that MOVE remains reliable even under early-stage miscalibration.
>
> |Method|$\mathbf{n}_0$|Acc. ratio|Cost ratio|
> |-|-|-|-|
> |LOC|5%|-0.05|0.0292|
> |LOC|10%|0.03|0.0484|
> |MOVE|5%|0.07|0.1224|
> |MOVE|10%|0.02|0.0423|
>
> ## 6.	Robustness of MOVE to Violations of the Monotonicity Assumption [Q3]
>
> The training observations used in our experiments already contain some monotonicity violations. To quantify this, we measure the monotonicity violation rate: the fraction of adjacent input pairs where performance does not increase, which is 23.1% for CIFAR-100. Despite this, MOVE consistently outperforms baselines for most settings, demonstrating robustness to local non-monotonicity in the training data.
>
> ## 7.	Limitation of Coordinate-wise Monotonicity in Capturing Source Interaction Effects [Q4]
>
> We agree that coordinate-wise monotonicity does not explicitly enforce interaction structure. However, the GP with an RBF kernel is defined over the joint input space and can capture non-additive interactions through its covariance function. To examine this empirically, we analyze the marginal effect of increasing $n_5$ from 1250 to 1500 at two different values of $n_1$ using MOVE trained on CIFAR-10. The differing marginal effects confirm that MOVE captures interaction effects between sources.
>
> |$n_1$|$n_5=1250$|$n_5=1500$|$\delta$|
> |---|---|---|---|
> |1000|78.47|79.29|+0.82|
> |1500|85.37|86.83|+1.46|

---

> > ### Author Rebuttal · Reviewer_HcL9 · 2026-04-05
> >
> > Thank you for the response, I am happy with the additional experimental validation and clarifications provided. I will raise my score 4 -> 5.

---

> > > ### Author Response · Authors · 2026-04-06
> > >
> > > We sincerely appreciate your careful reading of our rebuttal and the subsequent adjustment to our score. We are glad the clarifications helped you better understand our methods and results. We will be sure to include the above points in our revision to improve and clarify our paper.

---

### Official Review · Reviewer_ZcBG · 2026-03-13

**Soundness:** 3
**Presentation:** 3
**Significance:** 2
**Originality:** 3
**Overall Recommendation:** 4
**Confidence:** 4

**Summary:**

This paper proposes a method for multi-source data collection that combines a monotonic variational Gaussian process with an expected-shortfall acquisition strategy.
The monotonic variational GP models the learning curve as a function of per-source sample sizes, enforcing coordinate-wise monotonicity via virtual derivative constraints through a probit likelihood.
The expected-shortfall acquisition strategy replaces the failure probability penalty used in prior work, which the authors show suffers from vanishing gradients when performance is far below the target.
Experiments on CIFAR-10, CIFAR-100, VOC segmentation, and VOC object detection show that the proposed method achieves comparable accuracy ratios to the baseline while substantially reducing cost ratios.

**Compliance With Llm Reviewing Policy:**

Affirmed.

**Final Justification:**

My position still leans toward acceptance, and my score reflects the current state of the paper.

**Key Questions For Authors:**

1. I wonder how well the monotonicity constraint is satisfied in practice.
It would be great to have some plots showing the learned learning curves and how they compare to the true curves.

1. In the first term of Eq (5), why does it multiply the probability of failure \\(1 - F\\) with the number of data collected in each round (\\(n_t - n_{t - 1}\\))?
Similarly, why including \\(R(n_{t - 1})\\) in Eq (29)?
Is it necessary?

**Limitations:**

No societal concerns.

**Strengths And Weaknesses:**

1. The monotonic variational GP formulation using virtual derivative constraints (Riihimaki & Vehtari, 2010) is a sensible modeling choice.
The idea of reducing monotonicity constraints to classification problems via the probit likelihood is intuitive.
However, monotonicity is only softly encouraged rather than strictly enforced, acting as a regularizer whose strength depends on the weight \\(\lambda\\).
Additionally, the probit likelihood is non-Gaussian, and in my experience, training variational GPs with the probit likelihood is substantially slower than variational GP regression alone.
This computational overhead is not discussed in the paper.

1. Replacing the failure probability penalty with expected shortfall is well-motivated.
The intuition mirrors the well-known advantage of expected improvement over probability of improvement in Bayesian optimization.
The authors further justify this with an analysis showing that the failure probability suffers vanishing gradient issues.

1. The experimental setup is somewhat contrived.
The multi-source setting is constructed by partitioning class labels into groups (e.g., 5 groups of 2 classes each for CIFAR-10), rather than using the raw labels as individual sources.
It is unclear why \\(K = 10\\) sources (one per class) was not used directly.
Is this a limitation of the GP model, which places virtual derivative constraints at \\(J\\) locations per coordinate and thus scales as \\(J \cdot K\\) in the number of auxiliary variables?
If so, this scalability limitation in \\(K\\) should be discussed.

1. Lack of baselines.
The learning curve ablation experiments only compares with variational GP models (e.g., SVGP and DSVGP).
For completeness, it would be great to check how exact GPs (with no monotonicity constraints at all) would perform here.

**Minors**
1. Line 302: when -> When

---

> ### Author Rebuttal · Authors · 2026-03-31
>
> We thank the reviewer for their detailed review and their insightful questions. Please find our response below. (W for weakness, Q for Questions)
>
> ## 1.	Soft Monotonicity Constraints and Computational Overhead [W1]
>
> We chose soft monotonicity to improve robustness under noisy observations, while acknowledging that it does not provide a hard guarantee. To assess sensitivity to $\lambda$, we conduct an ablation in the same setting as Figure 3. The results suggest that performance is fairly stable across the tested $\lambda$ values in NLL.
>
> |Model|$\lambda$|MSE|NLL|
> |---|---|---|---|
> |MOVE|0.4|16.51|3.21|
> |MOVE|0.5|10.45|3.20|
> |MOVE|0.6|9.96|3.16|
>
> Regarding the computational overhead of the probit likelihood, we acknowledge that MOVE incurs additional GP fitting time compared to SVGP and DSVGP due to the probit likelihood terms, as shown below.
>
> |Model|Training time|Inference time|Memory|
> |---|---|---|---|
> |Exact GP|15s|0.01s|112MB|
> |SVGP|107s|0.02s|20MB|
> |DSVGP|262s|0.04s|54MB|
> |MOVE|429s|0.06s|110MB|
>
> While MOVE requires additional GP fitting time, our results indicate improved predictive quality and more cost-efficient data collection decisions. Since the primary goal is to reduce total data collection cost, this additional fitting overhead should be weighed against the downstream savings, and in our main experiments, the tradeoff is favorable.
>
> ## 2.	Experimental Setup and Scalability with Respect to K [W2]
>
> Scalability with respect to K involves both the cost of generating learning curve observations and the surrogate model complexity. Each observation $V(\mathbf{n})$ requires fully training a downstream model; covering a K-source grid with L budget levels requires $O(L^K)$ training runs. For instance, in CIFAR-100 experiments, with $L=3$ and $K=10$, this would require over 59,000 ResNet18 training runs. This bottleneck is inherent to all learning curve based methods. For context, a strong baseline LOC (Mahmood et al., NeurIPS 2022) was evaluated only on $K=1$ and $K=2$ sources, leaving larger combinatorial settings largely unexplored. Our current experiments extend this setting to $K=5$. To provide additional context, we include $K=2$ experiments in Appendix C.2. These results suggest that the method remains effective across different source configurations. Regarding scalability, the number of auxiliary variables scales as $J$$\cdot$$K$. Since $J$ is a tunable hyperparameter, one possible mitigation is to use a smaller $J$ as $K$ increases. To examine this tradeoff, we conduct an additional $J$ ablation.
>
> |Model|J|MSE|NLL|
> |---|---|---|---|
> |MOVE|2|12.42|3.17|
> |MOVE|3|4.03|2.93|
> |MOVE|4|4.13|2.70|
>
> ## 3.	Lack of Baselines [W3]
>
> We appreciate the suggestion. We have now added Exact GP as an additional baseline. We report results for two settings of MOVE: the minimal monotonicity regularization setting and a stronger setting. Even with minimal monotonicity constraints, MOVE outperforms all baselines, which is consistent with the virtual pseudo-observations improving extrapolation beyond the observed range.
>
> |Model|MSE|NLL|
> |---|---|---|
> |Exact GP|4358.60|14.25|
> |SVGP|1127.60|12.01|
> |DSVGP|1240.50|12.09|
> |MOVE ($\lambda=0.001$)|700.99|7.68|
> |MOVE ($\lambda=0.5$)|10.45|3.20|
>
> ## 4.	Visualization of Learned Learning Curves and Monotonicity [Q1]
>
> We fully agree with that question. We provide visualizations of the learned learning curves alongside the true performance values. Since direct visualization of a 5-dimensional learning curve is infeasible, we use the K=2 CIFAR-10 training set from Appendix C.2, allowing marginal curve comparison per source. The plots are available at the following anonymous link:
>
> https://drive.proton.me/urls/41PZVKAJVW#nyNo5XABlGtq
>
> As shown, the learned curve remains monotone while closely tracking the overall trend of the true performance values, confirming that the virtual derivative constraints effectively enforce the desired monotonic structure in practice.
>
> ## 5.	Role of the Failure Probability Term in the Objective [Q2]
>
> The role of $(1 - F(\mathbf{n}_{t-1}))$ is to prevent the objective from collapsing into a terminal-only formulation. Without this term, the objective reduces to $\mathbf{c}^\top(\mathbf{n}\_T - \mathbf{n}\_0) + \gamma(1 - F(\mathbf{n}\_T))$
> which depends only on the final allocation $\mathbf{n}\_T$, making intermediate decisions irrelevant. In MOVE, the model is updated at each round, but without $1 - F(\mathbf{n}\_{t-1})$, this sequential feedback is not reflected in the objective, and the optimization effectively ignores the remaining time horizon, potentially allocating resources inefficiently across rounds.
>
> By contrast, weighting each stage by $1 - F(\mathbf{n}\_{t-1})$ makes the objective responsive to the current success probability, so that intermediate decisions remain relevant in the multi-stage setting. The same role applies to $R(\mathbf{n}\_{t-1})$ in Eq. (29).

---

> > ### Author Rebuttal · Reviewer_ZcBG · 2026-04-03
> >
> > > We appreciate the suggestion. We have now added Exact GP as an additional baseline. We report results for two settings of MOVE: the minimal monotonicity regularization setting and a stronger setting. Even with minimal monotonicity constraints, MOVE outperforms all baselines, which is consistent with the virtual pseudo-observations improving extrapolation beyond the observed range.
> >
> > Do the authors have explanations why the exact GP produces much worse MSE and NLL compared to variational GPs? This is quite counter intuitive.

---

> > > ### Author Response · Authors · 2026-04-06
> > >
> > > >**We thank the reviewers for the thoughtful comments.  Regarding the additional question, we have provided the following clarifications.**
> > > ---
> > >
> > > Exact GP and SVGP/DSVGP optimize fundamentally different objectives. Exact GP maximizes the marginal likelihood
> > >
> > > $$\log p\_\theta (\mathbf{y}\mid X)=-\frac{1}{2}\mathbf{y}^\top (K^{\theta}\_{XX} + \sigma^2 I)^{-1}\mathbf{y}-\frac{1}{2}\log \lvert K^{\theta}\_{XX} + \sigma^2 I \rvert + \text{const},$$
> > >
> > > whereas SVGP optimizes a variational evidence lower bound (ELBO)
> > >
> > > $$\mathcal{L}\_{\text{SVGP}} = \sum\_i \mathbb{E}\_{q(f\_i)}[\log p(y\_i \mid f\_i)] - \mathrm{KL}(q(\mathbf{u}) \| p(\mathbf{u})).$$
> > >
> > > This corresponds to finding the closest approximation to the true posterior within a restricted variational family. We would also like to clarify that SVGP/DSVGP is not a computational approximation to Exact GP, but a distinct variational inference framework with its own principled objective over a restricted posterior family. This distinction becomes particularly apparent in extrapolation behavior.
> > >
> > > While Exact GP interpolates well within the training range, its extrapolation is determined by the learned hyperparameters. The ELBO includes a KL term that explicitly penalizes the variational distribution from deviating from the GP prior over the inducing variables, which provides an explicit form of complexity control, in contrast to the implicit complexity control induced by the marginal likelihood in Exact GP. This difference in how complexity is controlled during optimization may lead to different extrapolation behavior, particularly under limited data or model misspecification.
> > >
> > > We provide predictive curve visualizations at https://drive.proton.me/urls/0BKMSVB5P0#8NNcoDLLpSFE
> > >
> > > which empirically show that variational approaches tend to produce smoother and more conservative predictions outside the observed data range, consistent with the observed differences in extrapolation MSE and NLL in our experiments.
> > >
> > > ---
> > > >**We thank you again for your detailed feedback. We believe these updates will significantly improve the presentation of our results and provide the necessary confidence in our findings.**

---

### Decision · Program_Chairs · 2026-04-30

**Decision:**

Accept (regular)

**Comment:**

This paper tackles an important problem, and the reviewers generally agree that the method is technically sound and well motivated. The discussion centered mainly on the level of novelty and on the breadth and scalability in the empirical validation. The rebuttal addressed most of the earlier concerns by adding baselines, a hyperparameter sensitivity analysis, runtime details (wall-clock time and memory), and larger-scale results on ImageNet.

While some concerns remain about novelty and empirical scope, I believe the paper is technically solid and will be of interest to at least part of the community, so I recommend acceptance.